# Incorporating Expert Priors into Bayesian Optimization via Dynamic Mean Decay

**Chongqi Qu**[1], **Meiqin Liu**[1,2,*], **Jian Lan**[3], **Shanling Dong**[2], **Zhunga Liu**[4]

[1]National Key Laboratory of Human-Machine Hybrid Augmented Intelligence,
Xi'an Jiaotong University, Xi'an 710049, China
[2]College of Electrical Engineering, Zhejiang University,
Hangzhou 310027, China
[3]School of Electronics and Information Engineering,
Xi'an Jiaotong University, Xi'an 710049, China
[4]School of Automation, Northwestern Polytechnical University,
Xi'an 710072, China
`QuChongqi@stu.xjtu.edu.cn`, `{liumeiqin,shanlingdong28}@zju.edu.cn`
`lanjian@mail.xjtu.edu.cn`, `liuzhunga@nwpu.edu.cn`

## Abstract

Bayesian optimization (BO) is a powerful approach for black-box optimization, and in many real-world problems, domain experts possess valuable prior knowledge about promising regions of the search space. However, existing prior-informed BO methods are often overly complex, tied to specific acquisition functions, or highly sensitive to inaccurate priors. We propose `DynMeanBO`, a simple and general framework that incorporates expert priors into the Gaussian process mean function with a dynamic decay mechanism. This design allows BO to exploit expert knowledge in the early stages while gradually reverting to standard BO behavior, ensuring robustness against misleading priors while retaining the exploratory behavior of standard BO. `DynMeanBO` is broadly compatible with acquisition functions, introduces negligible computational cost, and comes with convergence guarantees under Expected Improvement and Upper Confidence Bound. Experiments on synthetic benchmarks and hyperparameter optimization tasks show that `DynMeanBO` accelerates convergence with informative priors and remains robust under biased ones.

## 1 Introduction

Black-box optimization aims to optimize objective functions that are expensive to evaluate and lack analytical expressions or gradient information. Among various approaches, Bayesian optimization (BO) (Jones et al., 1998; Shahriari et al., 2015; Frazier, 2018; Garnett, 2023) has emerged as a powerful and sample-efficient framework by constructing a probabilistic surrogate model of the objective function and selecting promising candidates via an acquisition function. BO has found widespread applications in hyperparameter optimization (HPO) (Bergstra et al., 2011; Snoek et al., 2012), automated materials discovery (Li et al., 2017; Zhang et al., 2020), and robotics control optimization (Antonova et al., 2017; Calandra et al., 2016).

While BO has achieved remarkable success, fully automated optimization remains challenging in many real-world scenarios. In domains where evaluations are costly, practitioners often rely on prior knowledge to guide the search. For example, in HPO, experts frequently use heuristic rules or accumulated experience to identify promising hyperparameter regions rather than applying BO from scratch (Bouthillier & Varoquaux, 2020; Smith et al., 2018; He et al., 2019; Li et al., 2024a; Marek et al., 2025). Incorporating expert priors into BO can accelerate optimization by complementing BO's efficiency with expert insights. However, prior-informed BO methods (Ramachandran et al.,

---

*Corresponding author.

2020; Souza et al., 2021; AV et al., 2022; Hvarfner et al., 2022; Huang et al., 2023; Hvarfner et al., 2024) are often complex or difficult to generalize across acquisition functions.

In this paper, we propose Dynamic Mean Bayesian Optimization (`DynMeanBO`), a framework that incorporates expert priors into the Gaussian process(GP) mean function and gradually decays their influence as more data is collected. This design allows the optimizer to exploit prior knowledge in the early stages of exploration while mitigating potential bias from inaccurate priors in the long run.

Our main contributions are:

- We introduce `DynMeanBO`, a BO framework that integrates expert priors via a dynamically decaying mean function.
- The method is lightweight, broadly compatible with acquisition functions, and adds negligible computational overhead.
- We provide convergence guarantees for `DynMeanBO` under Expected Improvement (`EI`) and Upper Confidence Bound (`UCB`).
- Experiments on synthetic functions and HPO tasks show that `DynMeanBO` accelerates convergence with accurate expert priors, remains robust under biased expert priors, and consistently outperforms other prior-informed approaches.

## 2 BACKGROUND

**Notations.** Scalars are denoted by lowercase letters (e.g., $f$), vectors by bold lowercase letters (e.g., $\mathbf{x}$), and matrices by bold uppercase letters (e.g., $\mathbf{K}$). The search space is $\mathcal{X} \subseteq \mathbb{R}^d$, and the observed dataset of $n$ points is $\mathcal{D}_n = \{\mathbf{X}, \mathbf{y}\}$. We denote the GP posterior mean and variance as $\mu_n(\mathbf{x})$ and $s_n^2(\mathbf{x})$, respectively, and the observation noise variance as $\sigma^2$. The global optimum location and its value are denoted by $\mathbf{x}^*$ and $f(\mathbf{x}^*)$, expert priors over the optimum location are denoted as $\pi(\mathbf{x})$, acquisition functions are denoted as $\alpha(\mathbf{x})$ and $\mathbb{E}[\cdot]$ denotes expectation.

### 2.1 BAYESIAN OPTIMIZATION

BO is a framework for optimizing expensive black-box functions by sequentially selecting evaluation points. Given an unknown objective function $f$, the goal is to find its global maximizer:

$$\mathbf{x}^* = \arg\max_{\mathbf{x} \in \mathcal{X}} f(\mathbf{x}). \tag{1}$$

At iteration $n$, a point $\mathbf{x}_n$ is evaluated with noisy observation $y_n = f(\mathbf{x}_n) + \varepsilon_n$, $\varepsilon_n \sim \mathcal{N}(0, \sigma^2)$. Conditioned on the observed dataset $\mathcal{D}_n = \mathcal{D}_{n-1} \cup \{\mathbf{x}_n, y_n\}$, a probabilistic surrogate model defines the posterior $p(f \mid \mathcal{D}_n)$. We adopt a GP as the surrogate model, which naturally provides a posterior distribution for the objective function; alternatives such as random forests (Hutter et al., 2011) or Bayesian neural networks (Springenberg et al., 2016; Li et al., 2024b) can also be used. The next evaluation point is selected by maximizing an acquisition function $\alpha(\mathbf{x})$, which balances exploration and exploitation.

### 2.2 GAUSSIAN PROCESS

A GP (Williams & Rasmussen, 2006) places a distribution over functions, enabling Bayesian non-parametric regression with principled uncertainty estimation. It is fully specified by a mean function $m(\mathbf{x})$ ($m : \mathcal{X} \rightarrow \mathbb{R}$) and a covariance (kernel) function $k(\mathbf{x}, \mathbf{x}')$ ($k : \mathcal{X} \times \mathcal{X} \rightarrow \mathbb{R}$). The mean function $m(\mathbf{x})$ can take any form, though it is often set to zero for simplicity in standard BO. The kernel function $k(\mathbf{x}, \mathbf{x}')$ encodes correlations between any two inputs, with common choices including the squared exponential (SE) and Matérn kernels (Frazier, 2018). The unknown objective $f(\mathbf{x})$ is modeled as a GP prior, $f(\mathbf{x}) \sim \mathcal{GP}(m(\mathbf{x}), k(\mathbf{x}, \mathbf{x}'))$. Given dataset $\mathcal{D}_n = \{\mathbf{X}, \mathbf{y}\}$, where $\mathbf{X} = [\mathbf{x}_1, ..., \mathbf{x}_n]^\top$ and $\mathbf{y} = [y_1, ..., y_n]^\top$, the posterior is also a GP: $p(f \mid \mathcal{D}_n) = \mathcal{GP}(\mu_n(\mathbf{x}), k_n(\mathbf{x}, \mathbf{x}'))$, with

$$\begin{aligned} \mu_n(\mathbf{x}) &= m(\mathbf{x}) + \mathbf{k}_n(\mathbf{x})^\top [\mathbf{K}_n + \sigma^2 \mathbf{I}]^{-1} (\mathbf{y} - \mathbf{m}), \\ k_n(\mathbf{x}, \mathbf{x}') &= k(\mathbf{x}, \mathbf{x}') - \mathbf{k}_n(\mathbf{x})^\top [\mathbf{K}_n + \sigma^2 \mathbf{I}]^{-1} \mathbf{k}_n(\mathbf{x}'), \end{aligned} \tag{2}$$

where $\mathbf{K}_n$ is the $n \times n$ kernel matrix with entries $[\mathbf{K}_n]_{ij} = k(\mathbf{x}_i, \mathbf{x}_j)$ for $i, j \in \{1, \ldots, n\}$, $\mathbf{k}_n(\mathbf{x}) = [k(\mathbf{x}_1, \mathbf{x}), ..., k(\mathbf{x}_n, \mathbf{x})]^\top$, $\mathbf{m} = [m(\mathbf{x}_1), ..., m(\mathbf{x}_n)]^\top$ and $\mathbf{I}$ is the $n \times n$ identity matrix. The posterior variance is $s_n^2(\mathbf{x}) = k_n(\mathbf{x}, \mathbf{x})$.

Kernel hyperparameters $\boldsymbol{\theta}$ are typically learned by maximizing the marginal likelihood. For a GP with mean function $m$, the marginal likelihood is $p(\mathbf{y} \mid \mathbf{X}, \boldsymbol{\theta}) = \mathcal{N}\big(\mathbf{y}; \mathbf{m}, \mathbf{K}_n + \sigma^2 \mathbf{I}\big)$, which leads to the log marginal likelihood

$$\log p(\mathbf{y} \mid \mathbf{X}, \boldsymbol{\theta}) = -\tfrac{1}{2}(\mathbf{y} - \mathbf{m})^\top (\mathbf{K}_n + \sigma^2 \mathbf{I})^{-1}(\mathbf{y} - \mathbf{m}) - \tfrac{1}{2}\log\big|\mathbf{K}_n + \sigma^2 \mathbf{I}\big| - \tfrac{n}{2}\log(2\pi). \quad (3)$$

## 2.3 ACQUISITION FUNCTION

Acquisition functions (AFs) are utility functions that guide the selection of the next evaluation point in BO, trading off exploitation of high-value regions with exploration of uncertain regions (Wang et al., 2023). A wide variety of AFs have been proposed, each with distinct characteristics. Improvement-based methods include Probability of Improvement (PI) (Kushner, 1964), EI (Jones et al., 1998), and the Knowledge Gradient (KG) (Frazier et al., 2008), which aim to maximize the respective measures of improvement. Confidence-bound approaches include UCB (Srinivas et al., 2010); sampling-based methods include Thompson Sampling (TS) (Agrawal & Goyal, 2012); and information-theoretic strategies include Entropy Search (ES) (Hennig & Schuler, 2012), Predictive Entropy Search (PES) (Hernández-Lobato et al., 2014), and Max-value Entropy Search (MES) (Wang & Jegelka, 2017). Among these, EI and UCB are most widely used in practice.

**EI** maximizes the expected improvement over the current best observation $f_n^*$:

$$\begin{aligned} \alpha_{\mathrm{EI}}(\mathbf{x}) &= \mathbb{E}\left[\max\left(0, f(\mathbf{x}) - f_n^*\right)\right] \\ &= (\mu_n(\mathbf{x}) - f_n^*)\,\Phi\left(\frac{\mu_n(\mathbf{x}) - f_n^*}{s_n(\mathbf{x})}\right) + s_n(\mathbf{x})\,\phi\left(\frac{\mu_n(\mathbf{x}) - f_n^*}{s_n(\mathbf{x})}\right), \end{aligned} \quad (4)$$

where $\Phi(\cdot)$ and $\phi(\cdot)$ denote the cumulative distribution function and probability density function of the standard normal distribution, respectively.

**UCB** selects the next observation point based on the upper confidence bound of the predictive distribution. It balances exploration and exploitation as:

$$\alpha_{\mathrm{UCB}}(\mathbf{x}) = \mu_{n-1}(\mathbf{x}) + \beta_n^{1/2} s_{n-1}(\mathbf{x}), \quad (5)$$

where $\beta_n > 0$ is a parameter controlling the exploration–exploitation trade-off.

## 2.4 EXPERT PRIOR

In many domains, experts often possess prior knowledge about the likely location of the optimum $\mathbf{x}^*$ before evaluating a new task or model (Perrone et al., 2019). Such knowledge can be formalized as a probability distribution over the optimum location:

$$\pi(\mathbf{x}) = \mathbb{P}\left(\mathbf{x} = \arg\max_{\mathbf{x}' \in \mathcal{X}} f(\mathbf{x}')\right), \quad (6)$$

which encodes the likelihood that different inputs correspond to the global maximizer. In principle, the expert prior distribution $\pi(\mathbf{x})$ can take any form. In practice, the most commonly used distributions are Gaussian distributions, representing a single promising region, or mixtures of Gaussians, which can capture multiple promising regions in the search space. Figure 6 in Appendix A illustrates several examples of expert priors in one dimension, and the concept naturally extends to higher-dimensional search spaces.

Preference-based expert priors can also be constructed using a similar approach: when experts provide relative or pairwise preferences over candidate inputs, these preferences can be converted into a probability distribution over the optimum location. A mixture of Gaussians is often a convenient choice in this setting, as it can flexibly represent multiple favored regions implied by the expert preferences. In fact, expert priors can be easily converted into a probabilistic form, and this form is not limited to the Gaussian or Gaussian mixture examples used in the paper. Any probability distribution that can adequately express the expert's belief is valid. Further details can be found in Appendix A

## 3 RELATED WORK

Incorporating expert prior knowledge into BO has significant practical and theoretical value. Although the related literature remains limited, several representative studies have explored this direction. Nguyen & Osborne (2020) proposed a Bayesian framework for the scenario where the optimal function value $f^*$ is known, but its optimal location remains unknown. This approach works well when experts can precisely provide the optimal value; however, in most real-world tasks, such knowledge is rarely available, limiting its applicability. Huang et al. (2023) introduced the Preference Bayesian Neural Network (PBNN), which leverages a Siamese neural network to incorporate expert-provided preference feedback, effectively accelerating the BO process. Similarly, AV et al. (2022) proposed a human-in-the-loop BO framework, where human experts can directly intervene in the point selection process to improve search efficiency and performance.

The above methods incorporate expert priors in terms of the optimal value $f(\mathbf{x}^*)$, preference information, or feedback mechanisms. However, a more common and practical setting involves expert priors on the location of the optimum $\mathbf{x}^*$. BOPro (Souza et al., 2021) integrated expert priors into the BO-TPE framework (Bergstra et al., 2011), where experts design "good" and "bad" priors over $\mathbf{x}^*$, which are then combined within the BO-TPE structure to guide the optimization. However, this method cannot be applied to other more general BO approaches. Ramachandran et al. (2020) proposed a novel framework that directly embeds the cumulative distribution of expert priors over $\mathbf{x}^*$ into the kernel function. While interesting, this method is highly sensitive to inaccurate priors since they affect the entire kernel, potentially degrading performance significantly. Li et al. (2020) constructed a conditional posterior distribution incorporating expert priors over $\mathbf{x}^*$, defined as $p(\mathbf{x}^* \mid D_n, \pi) \propto p(\mathbf{x}^* \mid D_n)\pi(\mathbf{x}^*)$, and determined the next evaluation point by repeatedly sampling from this distribution. However, this approach can only be used with specific sampling strategies and cannot integrate with general acquisition functions, which restricts its flexibility. $\pi$BO (Hvarfner et al., 2022) integrates expert prior distributions into the acquisition function through a weighting mechanism, where the influence of the prior gradually decreases as more evaluations are performed. Although $\pi$BO is simple and effective and provides convergence guarantees for the EI acquisition function, it does not explicitly model the prior in the surrogate model and remains essentially heuristic. More recently, ColaBO (Hvarfner et al., 2024) is a highly flexible framework that injects expert priors as additional priors over the surrogate model, orthogonal to the traditional priors on kernel hyperparameters. While compatible with Monte Carlo (MC)-based acquisition functions, ColaBO cannot be used with non-MC acquisition functions and incurs substantial computational costs.

Despite these advances, existing methods still face several limitations: (1) some approaches are overly complex and difficult to implement in practice; (2) many rely on specific acquisition functions or sampling strategies, limiting their general applicability; and (3) several methods are highly sensitive to the quality of expert priors, which reduces robustness. Motivated by these challenges, we propose DynMeanBO, a simple yet effective framework that directly incorporates expert prior knowledge into the surrogate model by embedding it in the mean function of the GP. Unlike existing heuristic-based approaches, DynMeanBO achieves a principled integration of expert knowledge at the model level, is compatible with arbitrary acquisition functions, and demonstrates strong empirical performance across diverse benchmarks.

## 4 METHODOLOGY

We now present the proposed DynMeanBO framework. Unlike existing approaches that either embed priors heuristically or require specific acquisition functions, our method incorporates expert prior distributions on the location of the optimum by embedding them directly into the GP mean function. Although the design of DynMeanBO in this paper is implemented using a GP surrogate model, the method itself is not tied to any specific surrogate. It can equally be combined with other types of models, such as random forests (Hutter et al., 2011) or Bayesian neural networks (Springenberg et al., 2016; Li et al., 2024b). In Section 4.1, we construct a GP mean function based on expert priors. Section 4.2 details the overall algorithm framework, and Section 4.3 provides a theoretical analysis.

## 4.1 EXPERT-PRIOR-BASED MEAN FUNCTION

In BO, we adopt a GP surrogate model, $f(\mathbf{x}) \sim \mathcal{GP}(m(\mathbf{x}), k(\mathbf{x}, \mathbf{x}'))$, where $k(\mathbf{x}, \mathbf{x}') = \text{Cov}(f(\mathbf{x}), f(\mathbf{x}'))$ denotes the covariance and $m(\mathbf{x}) = \mathbb{E}[f(\mathbf{x})]$ encodes prior beliefs. In practice, $m(\mathbf{x})$ is often set to zero when the functional form of $f$ is unknown. When partial knowledge is available, for example if $f$ is approximately linear, a parametric mean of the form $m(\mathbf{x}) = a\mathbf{x} + b$ can be employed. The hyperparameters $a$ and $b$ are estimated from $\mathcal{D}_n$ via maximum likelihood, in the same way as kernel hyperparameters.

In most real-world problems, even domain experts rarely know the explicit form of $f(\mathbf{x})$. Instead, they may provide a prior distribution $\pi(\mathbf{x})$ over the likely location of the optimum $\mathbf{x}^*$. We incorporate this expert knowledge into the GP mean function as

$$m_{\text{prior}}(\mathbf{x}) = A \cdot \pi(\mathbf{x}) + B, \tag{7}$$

where $A > 0$ is a parameter controlling the scaling of $\pi(\mathbf{x})$, and $B$ is another parameter introducing an additive shift to $A\pi(\mathbf{x})$. Further details on the interpretation of $A$ and $B$, as well as their sensitivity analysis, can be found in Appendix B. Before any observations, $m_{\text{prior}}(\mathbf{x})$ reflects the shape of the expert prior, providing a coarse estimate of the function landscape consistent with prior beliefs about $\mathbf{x}^*$.

Figure 1 illustrates an example where a one-dimensional Gaussian prior $\pi(\mathbf{x})$ is incorporated into the GP mean function. Using the mean function $m_{\text{prior}}(\mathbf{x})$, which encodes expert prior knowledge, has a pronounced effect on both the prior and posterior distributions of $f$. Samples drawn from the prior $p_{\text{prior}}(f)$ and posterior $p_{\text{prior}}(f \mid \mathcal{D}_n)$ show a clear peak in the region deemed "good" by the expert (green area in Figure 1), indicating that the optimal values are highly likely to fall within this region. This confirms that the expert prior knowledge has been effectively integrated into the BO framework. For an example with a one-dimensional mixture of Gaussians as the expert prior, see Appendix C.

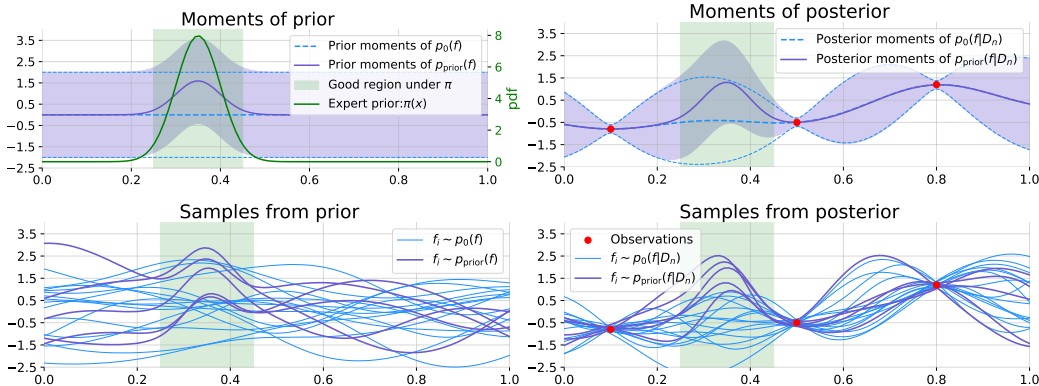

Figure 1: (Top left) Mean function and 95% confidence interval without any observations. $p(f)$ denotes the GP prior with $m(\mathbf{x}) = 0$, i.e., $p(f) = \mathcal{GP}(m(\mathbf{x}), k(\mathbf{x}, \mathbf{x}'))$, while $p_{\text{prior}}(f) = \mathcal{GP}(m_{\text{prior}}(\mathbf{x}), k(\mathbf{x}, \mathbf{x}'))$. (Top right) Mean function and 95% confidence interval conditioned on observed data $\mathcal{D}_n$. $p(f \mid \mathcal{D}_n)$ uses $m(\mathbf{x}) = 0$, whereas $p_{\text{prior}}(f \mid \mathcal{D}_n)$ uses $m_{\text{prior}}(\mathbf{x})$. (Bottom left) Samples drawn from the prior distributions $p(f)$ and $p_{\text{prior}}(f)$ without observations. (Bottom right) Samples drawn from the posterior distributions $p(f \mid \mathcal{D}_n)$ and $p_{\text{prior}}(f \mid \mathcal{D}_n)$ given $\mathcal{D}_n$.

## 4.2 BAYESIAN OPTIMIZATION WITH A DYNAMICALLY DECAYING MEAN FUNCTION

As more points are evaluated in BO, we gain increasing information about the objective function $f$, enabling a more accurate model of $f$. Thus, the reliance on the expert prior should progressively decline, which not only reflects the growing confidence in observed data but also preserves robustness in cases where the expert prior is far from the true optimum. Inspired by $\pi$BO (Hvarfner et al., 2022), we incorporate a decay mechanism into the mean function, defining the mean function at iteration $n$ after initialization as

$$m_n(\mathbf{x}) = \gamma_n \cdot m_{\text{prior}}(\mathbf{x}) + (1 - \gamma_n) \cdot \mu_0(\mathbf{x}), \quad \gamma_n = \exp(-\lambda(n - N_0)), \tag{8}$$

where $\lambda > 0$ controls the decay rate and $N_0$ is the number of initial evaluations. A sensitivity analysis of $\lambda$ is provided in Appendix J. The baseline mean function $\mu_0(\mathbf{x})$ corresponds to the mean function used in the standard BO setting, which is typically chosen as a constant function—most commonly the zero mean function. By gradually decaying the influence of the expert prior mean toward the baseline mean, the method remains robust even when the expert prior is substantially misaligned with the true optimum.

The complete procedure for incorporating expert prior knowledge into the BO framework is summarized in Algorithm 1. During initialization, we sample a portion of the initial points from the expert prior distribution $\pi(\mathbf{x})$, while the remaining points are drawn using Sobol sequences to ensure uniform coverage of the search space $\mathcal{X}$. This hybrid initialization strategy facilitates building a more accurate surrogate model and improves the subsequent optimization performance. Further analysis regarding the choice of the initialization ratio $\rho$ for sampling from $\pi(\mathbf{x})$ is presented in Appendix K.

---

**Algorithm 1 DynMeanBO**: Bayesian Optimization with Dynamic Mean Decay

---

**Require:** Search space $\mathcal{X}$, kernel function $k(\mathbf{x}, \mathbf{x}')$, expert prior distribution $\pi(\mathbf{x})$, decay coefficient $\lambda$, initial design size $N_0$, initialization ratio $\rho$, max iterations $N$, the parameters $A$ and $B$
**Ensure:** Optimized design $\mathbf{x}^*$

1: **Initialization:**
2: Draw $N_0^{(\text{prior})} = \lfloor \rho N_0 \rceil$ samples from $\pi(\mathbf{x})$
3: Draw $N_0^{(\text{Sobol})} = N_0 - N_0^{(\text{prior})}$ samples via Sobol sequences
4: Set $\{\mathbf{x}_i\}_{i=1}^{N_0} = \{\mathbf{x}_i\}_{i=1}^{N_0^{(\text{prior})}} \cup \{\mathbf{x}_j\}_{j=1}^{N_0^{(\text{Sobol})}}$
5: Observe $y_i = f(\mathbf{x}_i) + \varepsilon_i$, and set $\mathcal{D}_{N_0} = \{(\mathbf{x}_i, y_i)\}_{i=1}^{N_0}$
6: Initialize GP prior mean $m_{\text{prior}}(\mathbf{x}) = A \cdot \pi(\mathbf{x}) + B$
7: Fit GP posterior $p(f|\mathcal{D}_{N_0}) = \mathcal{GP}(\mu_{N_0}(\mathbf{x}), k_{N_0}(\mathbf{x}, \mathbf{x}'))$ according to Eq. (2)
8: **for** $n = N_0 + 1, \ldots, N$ **do**
9: $\quad$ $\mathbf{x}_n = \arg\max_{\mathbf{x} \in \mathcal{X}} \alpha(\mathbf{x}, \mathcal{D}_{n-1})$
10: $\quad$ $y_n = f(\mathbf{x}_n) + \varepsilon_n$
11: $\quad$ $\mathcal{D}_n = \mathcal{D}_{n-1} \cup \{(\mathbf{x}_n, y_n)\}$
12: $\quad$ Update mean function: $m_n(\mathbf{x})$ according to Eq.(8)
13: $\quad$ Update GP posterior: $p(f|\mathcal{D}_n) = \mathcal{GP}(\mu_n(\mathbf{x}), k_n(\mathbf{x}, \mathbf{x}'))$ according to Eq. (2)
14: **end for**
15: **return** $\mathbf{x}^* = \arg\max_{(\mathbf{x}_i, y_i) \in \mathcal{D}_N} y_i$

---

### 4.3 THEORETICAL ANALYSIS

We provide a theoretical analysis of the convergence properties of `DynMeanBO`, establishing guarantees for the commonly used `EI` and `UCB` acquisition functions. Complete proofs are presented in Appendix D and Appendix E. The analysis can be straightforwardly extended to other standard acquisition functions, following similar arguments; we omit these extensions for brevity.

**Convergence under `EI`.** To analyze the convergence of `DynMeanBO` under `EI`, we adopt the assumptions of Bull (2011). Although our focus is on *maximizing* the objective function, the theoretical framework of Bull (2011) assumes *minimization*. This distinction is immaterial, as maximization can be equivalently reformulated as minimization by considering $-f(\mathbf{x})$. Let $\mathcal{H}_k$ denote the reproducing kernel Hilbert space (RKHS) associated with a symmetric positive-definite kernel $k$. In our analysis, we employ the Matérn kernel (Matérn, 1960), where the smoothness of functions in $\mathcal{H}_k$ is controlled by the parameter $\nu$. We assume that the unknown objective function $f$ lies within a ball $B_R$ in $\mathcal{H}_k$, i.e., $\|f\|_{\mathcal{H}_k(\mathcal{X})} \leq R$.

We define the loss as

$$\mathcal{L}_n(u, \mathcal{D}_n, \mathcal{H}_k(\mathcal{X}), R) \triangleq \sup_{\|f\|_{\mathcal{H}_k(\mathcal{X})} \leq R} \mathbb{E}_f^u \big[ f(\mathbf{x}_n^*) - \min f \big], \tag{9}$$

where $u$ denotes the strategy, and $\mathbf{x}_n^*$ is the best point selected after $n$ evaluations. We denote the `EI` strategy under `DynMeanBO` as DynMeanBO-EI and the standard BO with `EI` strategy as BO-EI. Based on the detailed proof in Appendix D, we obtain the following theoretical result.

**Theorem 1** (Convergence of `DynMeanBO` under `EI`). *Let $\mathcal{X} \subset \mathbb{R}^d$ be compact, $f \in \mathcal{H}_k(\mathcal{X})$, and let `DynMeanBO` use the dynamic prior mean $m_n(\mathbf{x}) = \gamma_n \cdot m_{\mathrm{prior}}(\mathbf{x}) + (1 - \gamma_n) \cdot \mu_0(\mathbf{x})$ with $\gamma_n = \exp(-\lambda(n - N_0))$, $\lambda > 0$. Then, `DynMeanBO` under `EI` achieves the same asymptotic convergence rate as standard BO under `EI`, namely*

$$\mathcal{L}_n(\mathrm{DynMeanBO\text{-}EI}, \mathcal{D}_n, \mathcal{H}_k(\mathcal{X}), R) = O\big(n^{-(\nu \wedge 1)/d}(\log n)^\beta\big),$$

*where $\beta \geq 0$ is a constant depending on the kernel $k$ and $\nu$.*

**Convergence under `UCB`.** We establish the convergence of `DynMeanBO` under the `UCB` strategy by following the proof techniques of Srinivas et al. (2010). Our objective is to maximize the target function $f$, and at each iteration the next query point is chosen according to the `UCB` rule: $\mathbf{x}_n = \arg\max_{\mathbf{x} \in \mathcal{X}} \alpha_{\mathrm{UCB}}(\mathbf{x}) = \arg\max_{\mathbf{x} \in \mathcal{X}} \mu_{n-1}(\mathbf{x}) + \beta_n^{1/2} s_{n-1}(\mathbf{x})$. The resulting convergence guarantee is summarized below, with the complete proof provided in Appendix E.

**Theorem 2** (Convergence of `DynMeanBO` under UCB). *Let $\delta \in (0, 1)$. Assume that the true underlying function $f$ lies in the RKHS $\mathcal{H}_k$ associated with the kernel $k$, with $\|f\|_{\mathcal{H}_k}^2 \leq B$, and let $\beta_n = 2B + 300G_n \log^3(n/\delta)$. Assume further that the observational noise is $\sigma$-sub-Gaussian. Let $m_n(\mathbf{x}) = \gamma_n m_{\mathrm{prior}}(\mathbf{x}) + (1 - \gamma_n)\mu_0(\mathbf{x})$ be the dynamic prior mean of `DynMeanBO` at iteration $n$, where $\gamma_n \to 0$. When using the UCB acquisition function with parameters $\beta_n$, the cumulative regret of `DynMeanBO` satisfies, with probability at least $1 - \delta$,*

$$\Pr\left\{R_N \leq C_1\sqrt{N\beta_N G_N} + C_2 \sum_{n=1}^N \gamma_n \ \forall N \geq 1\right\} \geq 1 - \delta,$$

*where $R_N := \sum_{n=1}^N \big(f(\mathbf{x}^*) - f(\mathbf{x}_n)\big)$ denotes the cumulative regret, $G_N$ is the maximum information gain up to $N$, and $C_1, C_2 > 0$ are constants independent of $N$. In particular, if $\sum_{n=1}^\infty \gamma_n < \infty$, `DynMeanBO-UCB` achieves the same asymptotic convergence rate as `BO-UCB`:*

$$R_N = O\big(\sqrt{N\beta_N G_N}\big).$$

## 5 EXPERIMENTS

We systematically evaluate the performance of `DynMeanBO` on diverse tasks under both "good" (informative) and "bad" (misleading) expert priors. The experiments examine its compatibility with different acquisition functions and demonstrate its advantages over other prior-informed BO methods. Section 5.1 describes the experimental setup. Section 5.2 presents the compatibility results, while Section 5.3 provides the comparative analysis. Our implementation is publicly available at `https://github.com/quchongqi/DynMeanBO`

### 5.1 EXPERIMENTAL SETUP

**Expert Priors.** We adopt the expert prior settings used in $\pi$`BO` (Hvarfner et al., 2022) and `ColaBO` (Hvarfner et al., 2024), modeling both "good" and "bad" expert priors as Gaussian distributions. The mean of the "good" prior is located 10% away from the location of the global optimum, with variance set to 20% of the search space width. For the "bad" prior, the mean is shifted 70% away from the location of the global optimum; if this position lies outside the domain, it is clipped to the boundary, with the same variance setting. Detailed configurations are provided in Appendix F. To further investigate how different expert prior configurations influence the performance of `DynMeanBO`, we additionally study the effect of varying the prior variance under three prior-quality conditions: strong prior, weak prior, and wrong prior. The definitions and constructions of these prior types are provided in Appendix L, where we also present the corresponding detailed settings and analyses.

**Tasks.** We consider two types of tasks: synthetic functions and HPO benchmarks. The synthetic functions span 4D to 7D search spaces, including Hartmann (4D), Levy (5D), Hartmann (6D), Rosenbrock (6D), and Stybtang (7D) (Wang et al., 2020), all implemented in BoTorch[1] (Balandat et al., 2020). For HPO, we evaluate three 4D deep learning optimization problems from the

---

[1]`https://github.com/pytorch/botorch`

PD1 benchmark suite (WMT, CIFAR, and LM1B). Although their true optima are unknown, we leverage expert priors from MF-Prior-Bench[2] (Mallik et al., 2023). Moreover, we also examine higher-dimensional settings. In Appendix I, we report additional results on two 20-dimensional synthetic tasks — Levy (20D), Rosenbrock (20D).

**Comparison Algorithms.** To test the compatibility of `DynMeanBO` with different acquisition strategies, we evaluate it under seven widely used acquisition functions: `PI`, `EI`, `LogEI`, `TS`, `UCB`, `KG`, and `MES`. We compare the performance of standard BO and `DynMeanBO` under each acquisition function. Additionally, we benchmark `DynMeanBO` in comparison with the prior-informed BO methods, including $\pi$`BO` and `ColaBO`.

**Computational Platform.** All experiments are conducted on a dual-socket Intel Xeon Platinum 8575C server (2×48 cores, 192 threads, 2.80 GHz base / 4.00 GHz turbo, 4 NUMA nodes).

## 5.2 DYNMEANBO'S GENERALITY ACROSS ACQUISITION FUNCTIONS

To evaluate the generality of `DynMeanBO`, we test it with seven widely used acquisition strategies: `PI`, `EI`, `LogEI`, `TS`, `UCB`, `KG`, and `MES`. Since `DynMeanBO` incorporates expert priors directly into the GP mean function, it can be seamlessly combined with any acquisition function without requiring algorithmic modifications. We compare `DynMeanBO` against standard BO under all seven acquisition functions, and additionally include a random sampling baseline (`Sampling`) as a reference. For this experiment, a "good" expert prior is used to construct the `DynMeanBO` model.

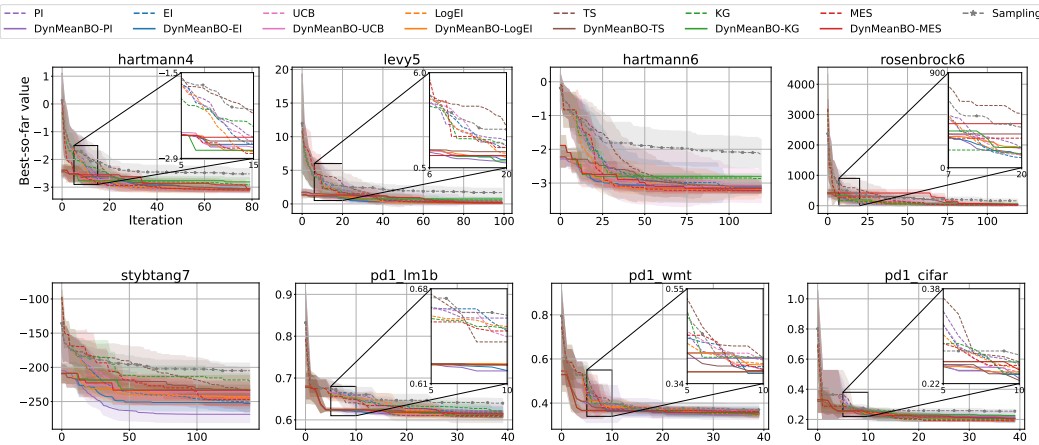

Figure 2: Performance on synthetic functions and HPO tasks. When a "good" expert prior is incorporated, `DynMeanBO` consistently finds better solutions faster than the standard BO across different acquisition functions.

The results in Figure 2 show that across all acquisition functions, `DynMeanBO` consistently accelerates convergence relative to standard BO. These improvements demonstrate its strong compatibility and effectiveness across a diverse set of strategies. In particular, during the early stages of optimization, `DynMeanBO` can leverage the expert prior to quickly identify better solutions, far more efficiently than standard BO. In practice, the tasks optimized with BO are often very expensive and time-consuming, so only a very limited number of evaluations can typically be performed. By incorporating expert prior knowledge, `DynMeanBO` is able to achieve substantial gains during these early stages of optimization, demonstrating even greater advantages in real-world scenarios.

We also compare the per-iteration evaluation time of `DynMeanBO` and standard BO, as shown in Figure 11 in Appendix G. The results indicate that the computational overhead of `DynMeanBO` is negligible. When provided with informative expert priors, `DynMeanBO` achieves faster convergence without sacrificing computational efficiency.

---

[2]https://github.com/automl/mf-prior-bench

## 5.3 COMPARATIVE STUDY OF DYNMEANBO AND EXISTING PRIOR-INFORMED BO FRAMEWORKS

We now compare `DynMeanBO` with other prior-informed BO methods. Under a "good" expert prior, `DynMeanBO` achieves performance comparable to $\pi$BO and `ColaBO`, while requiring lower computational cost. Under a "bad" expert prior, `DynMeanBO` exhibits strong robustness, maintaining stable performance even in the presence of misleading prior information.

**"Good" expert prior.** $\pi$BO and `ColaBO` are representative approaches that incorporate expert priors into BO. While $\pi$BO employs `EI`, `ColaBO` utilizes `LogEI` and `MES`, denoted as `MCpi-LogEI` and `MCpi-MES` in the figures. For a fair comparison, we evaluate `DynMeanBO` using the same acquisition functions—`EI`, `LogEI`, and `MES`. As shown in Figure 3, `DynMeanBO` achieves performance on par with $\pi$BO and `ColaBO`, while consistently accelerating the optimization process across all benchmarks.

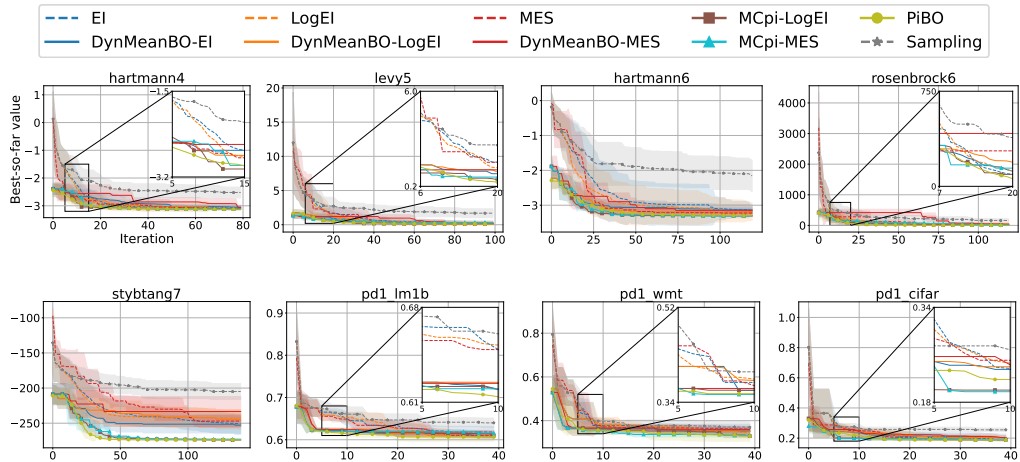

Figure 3: Performance on synthetic functions and HPO tasks under a "good" expert prior. `DynMeanBO`, $\pi$BO, and `ColaBO` achieve comparable results.

As discussed in Section 5.2, `DynMeanBO` introduces negligible computational overhead compared to standard BO. Furthermore, when compared with other prior-informed BO frameworks, `DynMeanBO` achieves substantially better computational efficiency. Figure 4 reports per-iteration evaluation time under identical acquisition functions, where `DynMeanBO` is markedly faster than both $\pi$BO and `ColaBO`.

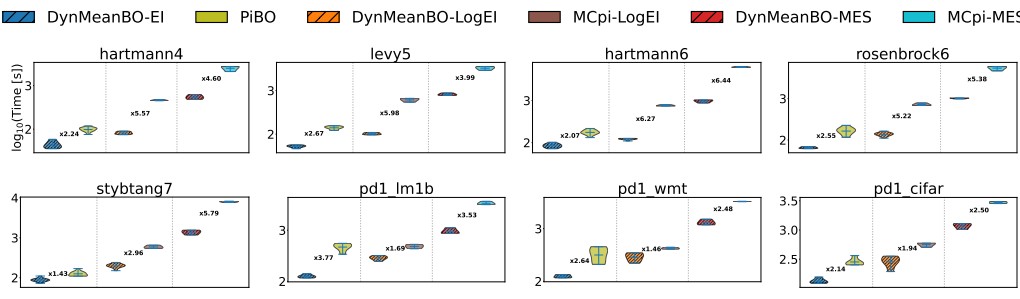

Figure 4: Per-iteration evaluation time ($\log_{10}$ scale) of `DynMeanBO`, $\pi$BO, and `ColaBO` on synthetic functions and HPO tasks under the "good" expert prior setting.

**"Bad" expert prior.** In practice, domain experts may provide priors that deviate substantially from the true optimum. The setup for the "bad" prior is detailed in Section 5.1. As shown in Fig. 5, `DynMeanBO` remains highly robust in this scenario, quickly approaching the performance of standard BO even when guided by an inaccurate prior. In terms of robustness, `DynMeanBO` clearly

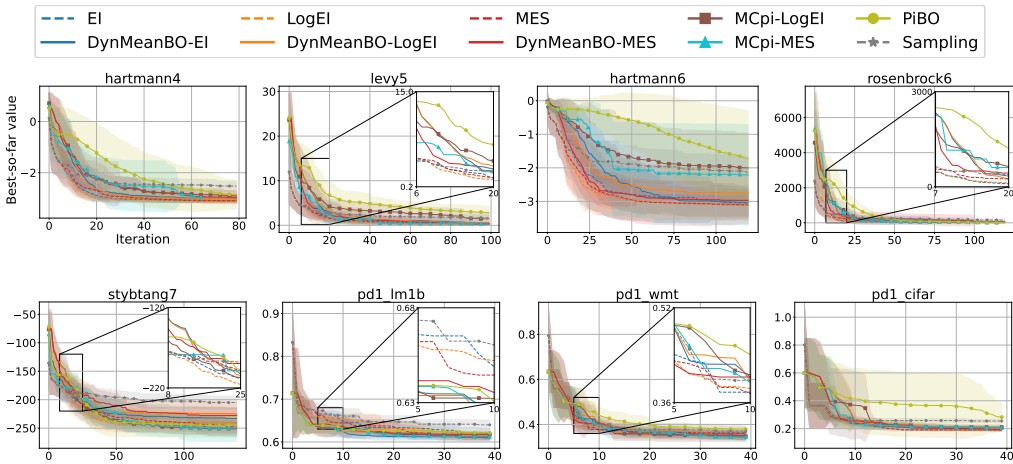

Figure 5: Performance on synthetic functions and HPO tasks under a "bad" expert prior. `DynMeanBO` demonstrates strong robustness.

outperforms both `ColaBO` and $\pi$`BO`, with a particularly large margin over $\pi$`BO`. Interestingly, on the PD1 (LM1B) task, all prior-informed methods (`DynMeanBO`, $\pi$`BO`, and `ColaBO`) converge faster than vanilla BO despite the misleading prior. This occurs because the "bad" prior, although far from the global optimum, still points to a high-quality suboptimal region. A comparison of computational overhead under the "bad" prior is provided in Appendix H.

## 6 CONCLUSION

We proposed `DynMeanBO`, a BO framework that incorporates expert prior knowledge via a dynamically decaying mean function. Our approach is compatible with any acquisition function and introduces negligible computational overhead. Empirically, `DynMeanBO` accelerates convergence when expert priors are informative, while remaining robust when priors are inaccurate. These findings demonstrate the practical benefit of integrating expert knowledge into BO and show that dynamically modulating the influence of the expert prior can effectively balance expert guidance with data-driven exploration.

For future work, we plan to combine `DynMeanBO` with complementary techniques, such as multi-fidelity optimization and parallel evaluation strategies. This combined approach aims to further accelerate the optimization process and enhance scalability, enabling applications to more complex models, diverse search spaces, and large-scale HPO tasks.

### ACKNOWLEDGMENTS

This research was supported by the National Natural Science Foundation of China under Grants U23B2035 and U24B20178.

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

## A    DETAILED EXPLANATION OF EXPERT PRIORS

To complement the discussion in Section 2.4, we provide visual illustrations of several representative expert prior distributions. These examples highlight how different choices encode domain knowledge about the likely location of the global optimum.

As shown in Figure 6, a unimodal Gaussian prior emphasizes a single promising region, while a Gaussian mixture prior flexibly represents multiple candidate regions. Preference-based priors can also be approximated using mixtures of Gaussians, which capture favored regions implied by expert preferences. Although we display one-dimensional cases for clarity, the same idea naturally extends to higher-dimensional spaces.

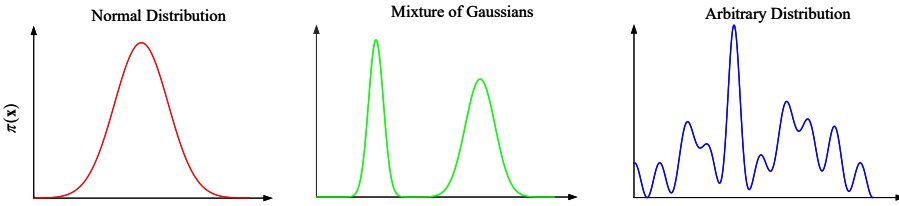

Figure 6: Illustration of different types of expert prior distributions.

Expert priors can naturally be expressed in the form of probability distributions. They are not limited to Gaussian or Gaussian mixture distributions; any distribution that captures the expert's belief can be used. For example, consider an arbitrary function $h(\mathbf{x})$, whose optimum reflects the expert's belief about where the optimal solution of the current task lies. We can normalize this function to obtain a expert prior distribution:

$$\pi(\mathbf{x}) = \frac{h(\mathbf{x})}{\int h(\mathbf{x})\,d\mathbf{x}}.$$

This allows us to incorporate expert knowledge into the optimization process through a probabilistic formulation.

## B    INTERPRETATION OF PARAMETERS $A$ AND $B$

We define the prior mean function as $m_{\mathrm{prior}}(\mathbf{x}) = A \cdot \pi(\mathbf{x}) + B$, where $\pi(\mathbf{x})$ denotes the expert prior distribution. This construction leverages expert knowledge to roughly shape the mean function, thereby highlighting the region where the function $f$ is most likely to achieve its maximum.

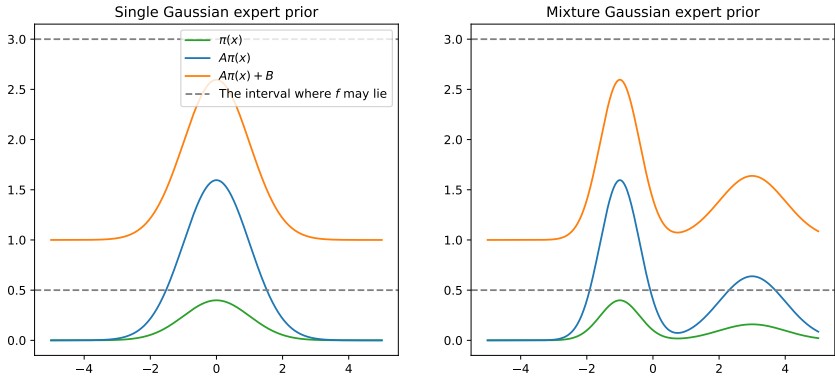

Figure 7: Illustration of the prior mean construction.

In this setup, $A$ serves as a scaling factor to align $\pi(\mathbf{x})$ with the magnitude of $f$, while $B$ acts as a shift that positions $A\pi(\mathbf{x})$ appropriately (e.g., around the mid-level of $f$). As illustrated in Figure 7,

the two subplots show how $A$ and $B$ influence the mean function when $\pi(\mathbf{x})$ is chosen as a Gaussian distribution and as a Gaussian mixture distribution, respectively.

The selection of $A$ and $B$ is flexible. They can be manually set, for example $A = 1$ and $B = 0$. Alternatively, they can be derived from the initial evaluations. Suppose in the initial set of evaluated points, the maximum and minimum observed values are $y_{\max}$ and $y_{\min}$. Then one may set

$$A = \frac{y_{\max} - y_{\min}}{\max_{\mathbf{x} \in \mathcal{X}} \pi(\mathbf{x})}, \quad B = \frac{y_{\max} + y_{\min}}{2}. \tag{10}$$

### B.1 SENSITIVITY ANALYSIS OF PARAMETERS A

In constructing our mean function, $m_{\mathrm{prior}}(\mathbf{x}) = A \cdot \pi(\mathbf{x}) + B$, the parameters $A$ and $B$ are primarily used to better approximate the true objective function. However, our main concern is the shape $\pi(\mathbf{x})$ of the mean function, as it is the key factor governing the balance between exploitation and exploration. In principle, the specific values of $A$ and $B$ should not have a major impact on the optimization process. To investigate the sensitivity of the algorithm to these parameters, we conduct a sensitivity analysis for both $A$ and $B$.

In this section, we focus on the sensitivity with respect to $A$. We consider $A = 0.1, 0.5, 1, 2, 10$ while fixing $B = 0$, and study how different values of $A$ affect the optimization process. We use `DynMeanBO-EI` as the test case, i.e., `DynMeanBO` with the EI acquisition function, to evaluate the sensitivity to $A$. The experimental results are shown in Figure 8.

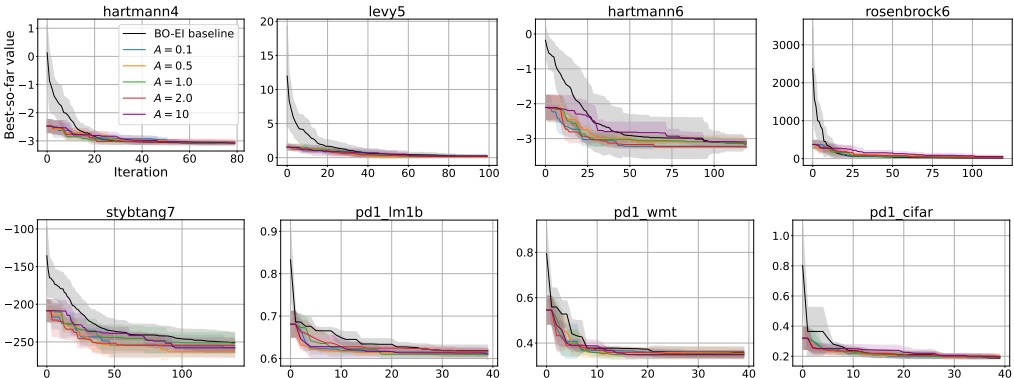

Figure 8: Ablation study of the parameter $A$ under the "good" expert prior setting

From the experimental results, we can see that the optimization process is largely insensitive to different values of $A$. In other words, the specific choice of $A$ is not critical; what matters most is the shape of the mean function.

### B.2 SENSITIVITY ANALYSIS OF PARAMETERS B

In this section, we analyze the sensitivity of the parameter $B$. Fixing $A = 1$, we examine how different values of $B = 0.0, 0.2, 0.4, 0.6, 0.8$ affect the optimization process within `DynMeanBO-EI`. The results are presented in Figure 9. From the experimental observations, the optimization process is highly insensitive to the choice of $B$, further confirming that the most critical factor is the shape of the mean function rather than the specific values of its scaling or shifting parameters.

## C  MIXTURE OF GAUSSIAN AS EXPERT PRIOR

Figure 10 illustrates an example where a one-dimensional mixture of Gaussians is used as the expert prior $\pi(\mathbf{x})$ and incorporated into the GP mean function. The mean function $m_{\mathrm{prior}}(\mathbf{x})$ now reflects multiple regions that the expert considers promising.

Samples drawn from the prior $p_{\mathrm{prior}}(f)$ and the posterior $p_{\mathrm{prior}}(f \mid \mathcal{D})$ show pronounced peaks in these regions (highlighted in green in Figure 10), indicating that the optimum is likely to lie within

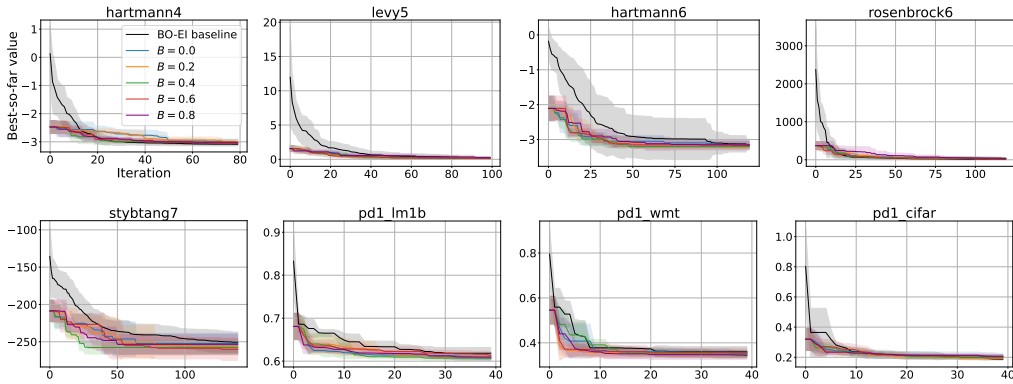

Figure 9: Ablation study of the parameter $B$ under the "good" expert prior setting

one or more of these areas. This demonstrates that, even when the expert prior takes the form of a Gaussian mixture model, it can still effectively encode multiple promising regions and significantly shape both the prior and posterior distributions of the objective function $f$ within the BO framework.

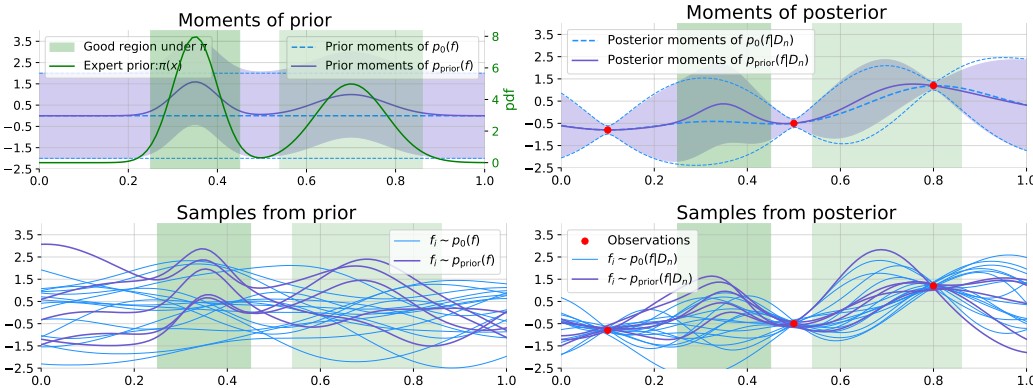

Figure 10: (Top left) Mean function and 95% confidence interval without any observations. $p(f)$ denotes the GP prior with $m(\mathbf{x}) = 0$, i.e., $p(f) = \mathcal{GP}(m(\mathbf{x}), k(\mathbf{x}, \mathbf{x}'))$, while $p_{\text{prior}}(f) = \mathcal{GP}(m_{\text{prior}}(\mathbf{x}), k(\mathbf{x}, \mathbf{x}'))$. (Top right) Mean function and 95% confidence interval conditioned on observed data $\mathcal{D}_n$. $p(f \mid \mathcal{D}_n)$ uses $m(\mathbf{x}) = 0$, whereas $p_{\text{prior}}(f \mid \mathcal{D}_n)$ uses $m_{\text{prior}}(\mathbf{x})$. (Bottom left) Samples drawn from the prior distributions $p(f)$ and $p_{\text{prior}}(f)$ without observations. (Bottom right) Samples drawn from the posterior distributions $p(f \mid \mathcal{D}_n)$ and $p_{\text{prior}}(f \mid \mathcal{D}_n)$ given $\mathcal{D}_n$.

## D  PROOF OF THEOREM 1

**Theorem 1** (Convergence of `DynMeanBO` under `EI`). *Let $\mathcal{X} \subset \mathbb{R}^d$ be compact, $f \in \mathcal{H}_k(\mathcal{X})$, and let `DynMeanBO` use the dynamic prior mean $m_n(\mathbf{x}) = \gamma_n \cdot m_{\text{prior}}(\mathbf{x}) + (1 - \gamma_n) \cdot \mu_0(\mathbf{x})$ with $\gamma_n = \exp(-\lambda(n - N_0))$, $\lambda > 0$. Then, `DynMeanBO` under `EI` achieves the same asymptotic convergence rate as standard BO under `EI`, namely*

$$\mathcal{L}_n(\text{DynMeanBO-EI}, \mathcal{D}_n, \mathcal{H}_k(\mathcal{X}), R) = O\big(n^{-(\nu \wedge 1)/d}(\log n)^{\beta}\big),$$

*where $\beta \geq 0$ is a constant depending on the kernel $k$ and $\nu$.*

*Proof.* The primary difference between `DynMeanBO` and standard BO lies in their mean functions. To analyze the convergence of `DynMeanBO` under `EI`, we first characterize the deviation of the mean function across iterations, which then allows us to study the convergence of the entire algorithm. To this end, we define the perturbation induced by the dynamic prior mean as

$$\delta_n(\mathbf{x}) := m_n(\mathbf{x}) - \mu_0(\mathbf{x}) = \gamma_n(m_{\text{prior}}(\mathbf{x}) - \mu_0(\mathbf{x})),$$

which captures the difference between the dynamic mean and the standard BO mean at iteration $n$. Since both $m_{\text{prior}}$ and $\mu_0$ are bounded, there exists $M > 0$ such that $\sup_{\mathbf{x} \in \mathcal{X}} |\delta_n(\mathbf{x})| \leq M \gamma_n$, which vanishes as $n \to \infty$.

The GP posterior mean is linear in the prior according to Eq. 2 , implying

$$
\begin{aligned}
\sup_{\mathbf{x} \in \mathcal{X}} |\mu_{\text{prior},n}(\mathbf{x}) - \mu_n(\mathbf{x})| &= \sup_{\mathbf{x} \in \mathcal{X}} |m_n(\mathbf{x}) - \mu_0(\mathbf{x}) + \mathbf{k}_n(\mathbf{x})^\top [\mathbf{K}_n + \sigma^2 \mathbf{I}]^{-1}(\mu_0 - \mathbf{m}_n)| \\
&= \sup_{\mathbf{x} \in \mathcal{X}} |\delta_n(\mathbf{x}) + \mathbf{k}_n(\mathbf{x})^\top [\mathbf{K}_n + \sigma^2 \mathbf{I}]^{-1} \delta_n(\mathbf{X})| \\
&= O(\sup_{\mathbf{x} \in \mathcal{X}} |\delta_n(\mathbf{x})|) \\
&= O(\gamma_n),
\end{aligned}
$$

where $\mu_{\text{prior},n}$ and $\mu_n$ denote the posterior means with dynamic prior $m_n(\mathbf{x})$ and base prior $\mu_0(\mathbf{x})$, respectively. And $\mu_0 = [\mu_0(\mathbf{x}_1), .., \mu_0(\mathbf{x}_n)]^\top$, $\mathbf{m}_n = [m_n(\mathbf{x}_1), .., m_n(\mathbf{x}_n)]^\top$.

According to the definition of EI in Equation 4, if we fix the posterior variance, then $\alpha_{\text{EI}}(\mathbf{x}) = g(u(\mathbf{x}))$, where $g(\cdot)$ is a function determined by the probability density function and the cumulative distribution function of the standard normal distribution. It follows that $g$ is differentiable and Lipschitz continuous. Therefore, there exists a constant $L > 0$ such that

$$
\begin{aligned}
\sup_{\mathbf{x} \in \mathcal{X}} |\alpha_{\text{DynMeanBO-EI}}(\mathbf{x}, n) - \alpha_{\text{BO-EI}}(\mathbf{x}, n)| &= \sup_{\mathbf{x} \in \mathcal{X}} |g(u_{\text{prior},n}(\mathbf{x})) - g(u_n(\mathbf{x}))| \\
&\leq L \cdot \sup_{\mathbf{x} \in \mathcal{X}} |u_{\text{prior},n}(\mathbf{x}) - u_n(\mathbf{x})| \\
&= O(\gamma_n).
\end{aligned}
$$

We define $\epsilon_n \triangleq \sup_{\mathbf{x} \in \mathcal{X}} |\alpha_{\text{DynMeanBO-EI}}(\mathbf{x}, n) - \alpha_{\text{BO-EI}}(\mathbf{x}, n)| = O(\gamma_n)$, and let $\mathbf{x}_{n+1} = \arg\max_{\mathbf{x} \in \mathcal{X}} \alpha_{\text{BO-EI}}(\mathbf{x}, n)$, $\mathbf{x}_{n+1}^{\text{prior}} = \arg\max_{\mathbf{x} \in \mathcal{X}} \alpha_{\text{DynMeanBO-EI}}(\mathbf{x}, n)$. Here, we make a reasonable assumption: there exist constants $c > 0$, $p \geq 1$, and $r > 0$ such that, for the maximizer

$$
\mathbf{x}^* = \arg\max_{\mathbf{x} \in \mathcal{X}} \alpha_{\text{BO-EI}}(\mathbf{x}, n)
$$

and any point $\mathbf{z}$ satisfying $\|\mathbf{z} - \mathbf{x}^*\| \leq r$, we have

$$
\alpha_{\text{BO-EI}}(\mathbf{x}^*, n) - \alpha_{\text{BO-EI}}(\mathbf{z}, n) \geq c \|\mathbf{x}^* - \mathbf{z}\|^p.
$$

Since $\mathbf{x}_{n+1} = \mathbf{x}^*$, we have

$$
\begin{aligned}
c\|\mathbf{x}_{n+1} - \mathbf{x}_{n+1}^{\text{prior}}\|^p &\leq \alpha_{\text{BO-EI}}(\mathbf{x}_{n+1}, n) - \alpha_{\text{BO-EI}}(\mathbf{x}_{n+1}^{\text{prior}}, n) \\
&= \alpha_{\text{BO-EI}}(\mathbf{x}_{n+1}, n) - \alpha_{\text{DynMeanBO-EI}}(\mathbf{x}_{n+1}, n) + \alpha_{\text{DynMeanBO-EI}}(\mathbf{x}_{n+1}, n) \\
&\quad - \alpha_{\text{BO-EI}}(\mathbf{x}_{n+1}^{\text{prior}}, n) \\
&\leq \alpha_{\text{BO-EI}}(\mathbf{x}_{n+1}, n) - \alpha_{\text{DynMeanBO-EI}}(\mathbf{x}_{n+1}, n) + \alpha_{\text{DynMeanBO-EI}}(\mathbf{x}_{n+1}^{\text{prior}}, n) \\
&\quad - \alpha_{\text{BO-EI}}(\mathbf{x}_{n+1}^{\text{prior}}, n) \\
&\leq |\alpha_{\text{BO-EI}}(\mathbf{x}_{n+1}, n) - \alpha_{\text{DynMeanBO-EI}}(\mathbf{x}_{n+1}, n)| + |\alpha_{\text{DynMeanBO-EI}}(\mathbf{x}_{n+1}^{\text{prior}}, n) \\
&\quad - \alpha_{\text{BO-EI}}(\mathbf{x}_{n+1}^{\text{prior}}, n)| \\
&\leq 2\epsilon_n
\end{aligned}
$$

So $\|\mathbf{x}_{n+1} - \mathbf{x}_{n+1}^{\text{prior}}\| \leq \left(\frac{2\epsilon_n}{c}\right)^{1/p}$, under the RKHS radius constraint $R$, all functions $f$ are Lipschitz continuous. Let $L_f$ denote a uniform Lipschitz constant. Then, we have

$$
|f(\mathbf{x}_{n+1}^{\text{prior}}) - f(\mathbf{x}_{n+1})| \leq L_f \cdot \|\mathbf{x}_{n+1} - \mathbf{x}_{n+1}^{\text{prior}}\| \leq L_f \cdot \left(\frac{2\epsilon_n}{c}\right)^{1/p}.
$$

Taking the supremum over both the expectation and the worst-case scenario, we obtain the following bound on the loss difference:

$$
\left| \mathcal{L}_n(\text{DynMeanBO-EI}, \mathcal{D}_n, \mathcal{H}_k(\mathcal{X}), R) - \mathcal{L}_n(\text{BO-EI}, \mathcal{D}_n, \mathcal{H}_k(\mathcal{X}), R) \right| \leq C \cdot \left(\frac{2\epsilon_n}{c}\right)^{1/p},
$$

where the constant $C$ depends on $L_f$ and the RKHS norm $R$ of the function.

Combining this with the known convergence rate of standard BO-EI (Bull, 2011):

$$\mathcal{L}_n(\text{BO-EI}, \mathcal{D}_n, \mathcal{H}_k(\mathcal{X}), R) = O\big(n^{-(\nu \wedge 1)/d}(\log n)^\beta\big),$$

where $\nu$ is the smoothness parameter of the kernel $k$ (Matérn kernel), and $\beta$ is defined as

$$\beta := \begin{cases} \alpha, & \text{if } \nu \leq 1, \\ 0, & \text{if } \nu > 1, \end{cases}$$

where $\alpha$ is the logarithmic correction exponent determined by the covering number of the RKHS. Since $\gamma_n = \exp\big(-\lambda(n - N_0)\big)$ decays exponentially, so

$$\mathcal{L}_n(\text{DynMeanBO-EI}, D_n, \mathcal{H}_k(\mathcal{X}), R) \leq \mathcal{L}_n(\text{BO-EI}, \mathcal{D}_n, \mathcal{H}_k(\mathcal{X}), R) + C \cdot \left(\frac{2\epsilon_n}{c}\right)^{1/p}$$

$$= O(\mathcal{L}_n(\text{BO-EI}, \mathcal{D}_n, \mathcal{H}_k(\mathcal{X}), R)) + O\left(C \cdot \left(\frac{2\epsilon_n}{c}\right)^{1/p}\right)$$

$$= O\big(n^{-(\nu \wedge 1)/d}(\log n)^\beta\big)$$

Therefore, `DynMeanBO` achieves the same asymptotic rate of convergence as standard BO. □

## E    PROOF OF THEOREM 2

**Theorem 2** (Convergence of `DynMeanBO` under `UCB`). *Let $\delta \in (0, 1)$. Assume that the true underlying function $f$ lies in the RKHS $\mathcal{H}_k$ associated with the kernel $k$, with $\|f\|^2_{\mathcal{H}_k} \leq B$, and let $\beta_n = 2B + 300G_n \log^3(n/\delta)$. Assume further that the observational noise is $\sigma$-sub-Gaussian. Let $m_n(\mathbf{x}) = \gamma_n m_{\text{prior}}(\mathbf{x}) + (1 - \gamma_n)\mu_0(\mathbf{x})$ be the dynamic prior mean of `DynMeanBO` at iteration $n$, where $\gamma_n \to 0$. When using the `UCB` acquisition function with parameters $\beta_n$, the cumulative regret of `DynMeanBO` satisfies, with probability at least $1 - \delta$,*

$$\Pr\left\{R_N \leq C_1\sqrt{N\beta_N G_N} + C_2\sum_{n=1}^{N}\gamma_N \; \forall N \geq 1\right\} \geq 1 - \delta,$$

*where $R_N := \sum_{n=1}^{N}\big(f(\mathbf{x}^*) - f(\mathbf{x}_n)\big)$ denotes the cumulative regret, $G_N$ is the maximum information gain up to $N$, and $C_1, C_2 > 0$ are constants independent of $N$. In particular, if $\sum_{n=1}^{\infty}\gamma_n < \infty$, `DynMeanBO` achieves the same asymptotic convergence rate as `BO-UCB`:*

$$R_N = O\big(\sqrt{N\beta_N G_N}\big).$$

*Proof.* We follow the analysis of BO-UCB by Srinivas et al. (2010), adapting it to account for the dynamic prior mean. Similar to the proof of Theorem 1, the key lies in analyzing the deviation between the mean functions of `DynMeanBO` and standard BO.

Following the proof of Theorem 1, the deviation between the mean functions of `DynMeanBO` and standard BO is given by

$$\Delta_n := \sup_{\mathbf{x} \in \mathcal{X}}\big|\mu_{\text{prior},n}(\mathbf{x}) - \mu_n(\mathbf{x})\big| = O(\gamma_n).$$

Therefore, there exists a constant $C_{\text{mean}} > 0$ such that $\Delta_n \leq C_{\text{mean}} \cdot \gamma_n$, which implies that the posterior mean perturbation vanishes as $\gamma_n \to 0$.

For baseline `BO-UCB`, Srinivas et al. (2010) show that, with probability at least $1 - \delta$,

$$|f(\mathbf{x}) - \mu_n(\mathbf{x})| \leq \sqrt{\beta_{n+1}}\, s_n(\mathbf{x}), \quad \forall \mathbf{x} \in \mathcal{X}.$$

(See Lemma 5.1 in Srinivas et al. (2010) for details.)

Combining this with the bound on $\Delta_n$ yields a high-probability confidence bound for `DynMeanBO`:

$$|f(\mathbf{x}) - \mu_{\text{prior},n}(\mathbf{x})| = |f(\mathbf{x}) - \mu_n(\mathbf{x}) + \mu_n(\mathbf{x}) - \mu_{\text{prior},n}(\mathbf{x})|$$

$$\leq |f(\mathbf{x}) - \mu_n(\mathbf{x})| + |\mu_n(\mathbf{x}) - \mu_{\text{prior},n}(\mathbf{x})|$$

$$\leq \sqrt{\beta_{n+1}}\, s_n(\mathbf{x}) + \Delta_n, \quad \forall \mathbf{x} \in \mathcal{X},$$

Here, let $\alpha_{\text{DynMeanBO-UCB}}$ denote the `UCB` acquisition function in `DynMeanBO`. According to Equation (5), we have $\alpha_{\text{DynMeanBO-UCB}}(\mathbf{x}, n) = \mu_{\text{prior},n}(\mathbf{x}) + \sqrt{\beta_{n+1}}\, s_n(\mathbf{x})$. Therefore, the function value satisfies

$$f(\mathbf{x}) \leq \alpha_{\text{DynMeanBO-UCB}}(\mathbf{x}, n) + \Delta_n.$$

Let $\mathbf{x}_n$ be the point chosen by `DynMeanBO` and $\mathbf{x}^* = \arg\max_{\mathbf{x}\in\mathcal{X}} f(\mathbf{x})$. Using the above confidence bound and the fact that $\mathbf{x}_n$ maximizes $\alpha_{\text{DynMeanBO-UCB}}(\mathbf{x}, n-1)$, the instantaneous regret satisfies

$$
\begin{aligned}
r_n &:= f(\mathbf{x}^*) - f(\mathbf{x}_n) \\
&\leq \alpha_{\text{DynMeanBO-UCB}}(\mathbf{x}^*, n-1) + \Delta_{n-1} - f(\mathbf{x}_n) \\
&\leq \alpha_{\text{DynMeanBO-UCB}}(\mathbf{x}_n, n-1) + \Delta_{n-1} - f(\mathbf{x}_n) \\
&= \mu_{\text{prior},n-1}(\mathbf{x}_n) + \sqrt{\beta_n}\, s_{n-1}(\mathbf{x}_n) + \Delta_{n-1} - f(\mathbf{x}_n) \\
&\leq |f(\mathbf{x}_n) - \mu_{\text{prior},n-1}(\mathbf{x}_n)| + \sqrt{\beta_n}\, s_{n-1}(\mathbf{x}_n) + \Delta_{n-1} \\
&\leq 2\big(\sqrt{\beta_n}\, s_{n-1}(\mathbf{x}_n) + \Delta_{n-1}\big) \\
&\leq 2\sqrt{\beta_n}\, s_{n-1}(\mathbf{x}_n) + 2C_{\text{mean}}\gamma_{n-1}.
\end{aligned}
$$

Summing over $n = 1$ to $N$ gives the cumulative regret

$$R_N := \sum_{n=1}^{N} r_n \leq 2 \sum_{n=1}^{N} \sqrt{\beta_n}\, s_{n-1}(\mathbf{x}_n) + 2C_{\text{mean}} \sum_{n=1}^{N} \gamma_{n-1}.$$

Next, we show that the cumulative regret is upper bounded. Let $\kappa := \sup_{\mathbf{x}\in\mathcal{X}} k(\mathbf{x}, \mathbf{x})$ (the kernel diagonal bound, finite since $\mathcal{X}$ is compact). For each term define $u_n := \sigma^{-2} s_{n-1}^2(\mathbf{x}_n)$; then $0 \leq u_n \leq U_{\max} := \kappa/\sigma^2$ and

$$
\begin{aligned}
G_N &:= \max_{\mathcal{D}\subset\mathcal{X}:|\mathcal{D}|=N} I(\mathbf{y}_{\mathcal{D}}; f_{\mathcal{D}}), \\
&= \max_{\mathcal{D}\subset\mathcal{X}:|\mathcal{D}|=N} \frac{1}{2} \log\big|\mathbf{I} + \sigma^{-2}\mathbf{K}_{\mathcal{D}}\big| \\
&\geq \frac{1}{2} \log\big|\mathbf{I} + \sigma^{-2}\mathbf{K}_{\mathcal{D}_N}\big| \\
&= \frac{1}{2} \log\left|\mathbf{I} + \sigma^{-2}\begin{bmatrix}\mathbf{K}_{\mathcal{D}_{N-1}} & \mathbf{k}_{N-1} \\ \mathbf{k}_{N-1}^\top & k(\mathbf{x}_n, \mathbf{x}_n)\end{bmatrix}\right| \\
&= \frac{1}{2} \log\left|\begin{bmatrix}\mathbf{I} + \sigma^{-2}\mathbf{K}_{\mathcal{D}_{N-1}} & \sigma^{-2}\mathbf{k}_{N-1} \\ \sigma^{-2}\mathbf{k}_{N-1}^\top & \mathbf{I} + \sigma^{-2}k(\mathbf{x}_n, \mathbf{x}_n)\end{bmatrix}\right| \\
&= \frac{1}{2} \log\{\big|\mathbf{I} + \sigma^{-2}\mathbf{K}_{\mathcal{D}_{N-1}}\big|\big(1 + \sigma^{-2}\big(k(\mathbf{x}_n, \mathbf{x}_n) - \mathbf{k}_{N-1}^\top(\mathbf{I} + \sigma^{-2}\mathbf{K}_{\mathcal{D}_{N-1}})^{-1}\mathbf{k}_{N-1}\big)\big)\} \\
&= \frac{1}{2} \log\{\big|\mathbf{I} + \sigma^{-2}\mathbf{K}_{\mathcal{D}_{N-1}}\big| \cdot \big(1 + \sigma^{-2}s_{N-1}^2(\mathbf{x}_N)\big)\} \\
&= \frac{1}{2} \sum_{n=1}^{N} \log(1 + \sigma^{-2}s_{n-1}^2(\mathbf{x}_n)) \\
&= \frac{1}{2} \sum_{n=1}^{N} \log(1 + u_n).
\end{aligned}
$$

Here, $I(\cdot)$ denotes the mutual information, and $G_N$ denotes the maximum information gain after $N$ steps; for its definition and computation, we refer the reader to Srinivas et al. (2010). The covariance matrix $\mathbf{K}_{\mathcal{D}_N}$ and the vector $\mathbf{k}_N$ are defined in Equation (2).

Consider the function $g(u) = \log(1 + u)/u$ for $u > 0$; $g$ is decreasing, so for all $u \in (0, U_{\max}]$,

$$\frac{\log(1 + u)}{u} \geq \frac{\log(1 + U_{\max})}{U_{\max}}.$$

It then follows that

$$u_n \leq \frac{U_{\max}}{\log(1 + U_{\max})} \log(1 + u_n),$$

and therefore

$$s_{n-1}^2(\mathbf{x}_n) = \sigma^2 u_n \le \frac{\sigma^2 U_{\max}}{\log(1 + U_{\max})} \log(1 + u_n) = \frac{\kappa}{\log(1 + \kappa/\sigma^2)} \log(1 + u_n).$$

Summing over $n$ and using $\sum_{n=1}^N \log(1 + u_n) = 2I(y_{1:N}; f_{1:N}) \le 2G_N$ yields

$$\sum_{n=1}^N s_{n-1}^2(\mathbf{x}_n) \le \frac{\kappa}{\log(1 + \kappa/\sigma^2)} \cdot 2G_N = C_0\, G_N,$$

where $I(y_{1:N}; f_{1:N})$ denotes the mutual information between the noisy observations $y_{1:N}$ and the latent function values $f_{1:N}$. And we set

$$C_0 := \frac{2\kappa}{\log(1 + \kappa/\sigma^2)}.$$

By Cauchy–Schwarz and monotonicity of $\{\beta_n\}$,

$$\sum_{n=1}^N \sqrt{\beta_n}\, s_{n-1}(\mathbf{x}_n) \le \sqrt{\Big(\sum_{n=1}^N \beta_n\Big)\Big(\sum_{n=1}^N s_{n-1}^2(\mathbf{x}_n)\Big)} \le \sqrt{N\beta_N \cdot C_0 G_N}.$$

Hence, for constants $C_1 := 2\sqrt{C_0}$ and $C_2 := 2C_{\mathrm{mean}}$, we obtain (with probability at least $1 - \delta$)

$$R_N \le C_1 \sqrt{N\beta_N G_N} + C_2 \sum_{n=1}^N \gamma_{n-1}.$$

In particular, if $\sum_{n=1}^\infty \gamma_n < \infty$ then the second term is bounded and we recover the asymptotic rate

$$R_N = O\big(\sqrt{N\beta_N G_N}\big),$$

which matches the `BO-UCB` rate.

$\square$

## F  SUPPLEMENTARY EXPERIMENTAL SETUP

For simplicity, we set the prior mean to $m_{\mathrm{prior}}(\mathbf{x}) = A \cdot \pi(\mathbf{x}) + B$ with $A = 1, B = 0$, use an initialization ratio of $\rho = 0.4$, and fix the decay factor in $\gamma_n = \exp(-\lambda(n - N_0))$ to $\lambda = 1$. All experiments employ the RBF kernel. For synthetic benchmarks, each experiment is repeated 10 times with different random seeds, whereas for HPO tasks, due to slower evaluation and higher computational cost, each experiment is repeated 5 times. The benchmarks and their respective settings are summarized in Table 1. All other parameters not explicitly specified are set to their default values in the BoTorch framework (Balandat et al., 2020).

## G  COMPUTATIONAL OVERHEAD UNDER DYNMEANBO AND STANDARD BO

As described in the `DynMeanBO` algorithm section, `DynMeanBO` integrates expert priors into the Gaussian process mean function in the form of a probabilistic distribution, effectively redesigning the BO mean function rather than introducing additional inference components. Therefore, this approach does not incur extra computational overhead. Theoretically, the computational complexity of `DynMeanBO` is nearly identical to that of standard BO, and our experimental results confirm this.

## H  COMPUTATIONAL OVERHEAD UNDER THE "BAD" EXPERT PRIOR

As shown in Figure 12, when a "bad" expert prior is used, the per-iteration evaluation time of `DynMeanBO` remains almost identical to that of standard BO under the same acquisition function, indicating that our dynamic mean adjustment introduces negligible additional overhead. In sharp contrast, both $\pi$BO and `ColaBO` are considerably slower due to their reliance on Monte Carlo sampling for incorporating expert prior information into the acquisition function. The computational

| Benchmark | Search space | $\mathbf{x}*$ | $N_0$ | $N$ |
|---|---|---|---|---|
| Hartmann (4D) | $[0,1]^4$ | $[0.19, 0.19, 0.56, 0.26]$ | 5 | 80 |
| Levy (5D) | $[-5,5]^5$ | $[1]^5$ | 6 | 100 |
| Hartmann (6D) | $[0,1]^6$ | $[0.20, 0.15, 0.48, 0.28, 0.31, 0.66]$ | 7 | 120 |
| Rosenbrock (6D) | $[-2.048, 2.048]^6$ | $[1]^6$ | 7 | 120 |
| Stybtang (7D) | $[-4,4]^7$ | $[-2.9]^7$ | 8 | 140 |
| PD1-WMT | $[0,1]^4$ | $[0.90, 0.69, 0.02, 0.97]$ | 5 | 40 |
| PD1-CIFAR | $[0,1]^4$ | $[1, 0.80, 0.0, 0.0]$ | 5 | 40 |
| PD1-LM1B | $[0,1]^4$ | $[0.91, 0.67, 0.36, 0.85]$ | 5 | 40 |
| Levy (20D) | $[-5,5]^{20}$ | $[1]^{20}$ | 8 | 140 |
| Rosenbrock (20D) | $[-2.048, 2.048]^{20}$ | $[1]^{20}$ | 8 | 140 |

Table 1: Experimental benchmarks used in our study. For each benchmark, we specify its search space, the location of the global optimum $\mathbf{x}^*$, the number of initial points ($N_0$), and the total evaluation budget ($N$).

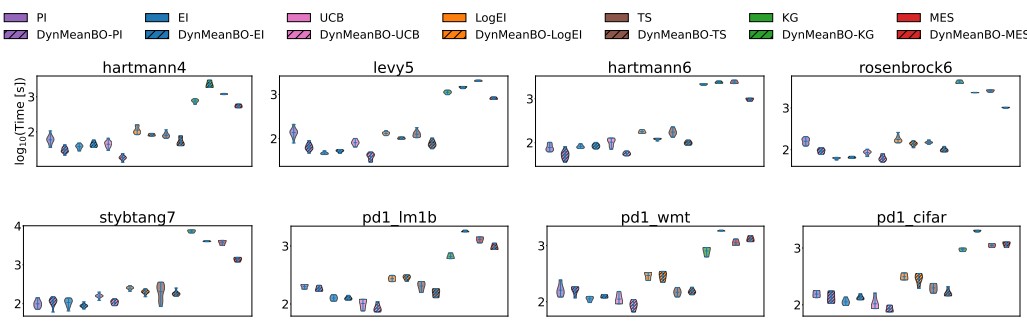

Figure 11: Per-iteration evaluation time ($\log_{10}$ scale) of DynMeanBO and standard BO on synthetic functions and HPO tasks.

burden is particularly pronounced for ColaBO, where employing MES necessitates nested Monte Carlo sampling to approximate the mutual information between candidate evaluations and the function maximum, leading to a dramatic runtime increase.

These results highlight that under inaccurate expert priors, $\pi$BO not only suffers from poor robustness but also incurs substantially higher computational cost. ColaBO alleviates some of these issues, showing improved robustness over $\pi$BO, but it still falls short of DynMeanBO and comes with considerably higher overhead. In contrast, DynMeanBO delivers both superior robustness and efficiency.

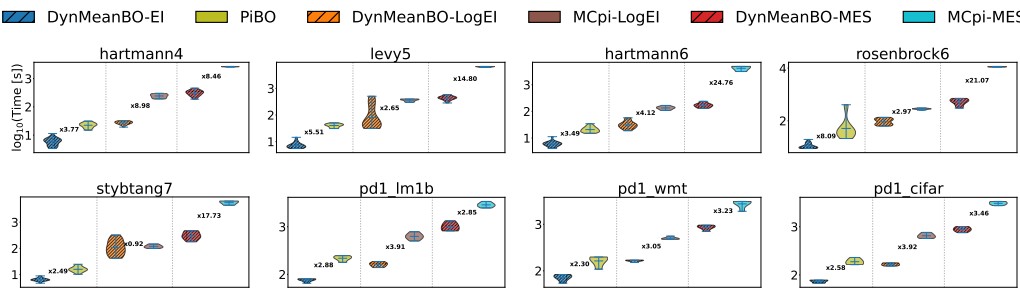

Figure 12: Per-iteration evaluation time ($\log_{10}$ scale) of DynMeanBO, $\pi$BO, and ColaBO on synthetic functions and HPO tasks under the "bad" expert prior setting.

# I  ADDITIONAL HIGH-DIMENSIONAL EXPERIMENTS

In addition to the eight commonly used benchmark tasks reported in the main text, we also evaluated DynMeanBO on higher-dimensional problems. Specifically, we included two 20-dimensional tasks—Levy (20D) and Rosenbrock (20D). The corresponding search space settings, optimum locations, number of initial points, total evaluation budget, and other relevant details are provided in Table 1.

First, we evaluate the performance of DynMeanBO on these two high-dimensional tasks under seven commonly used acquisition functions: PI, EI, LogEI, TS, UCB, KG, and MES. In this experiment, we use a "good" expert prior. The results are shown in Figure 13. Across all acquisition functions, DynMeanBO consistently accelerates convergence, demonstrating that it is not only seamlessly compatible with a wide range of acquisition strategies but also highly effective in high-dimensional settings.

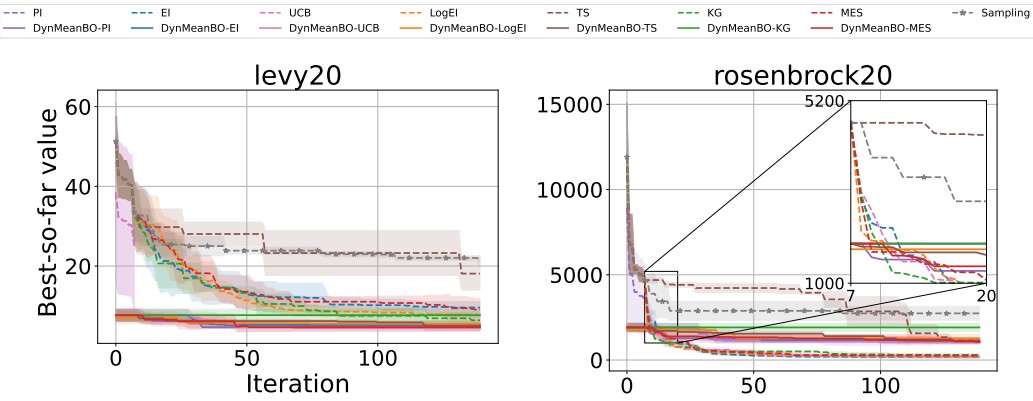

Figure 13: Performance on two high dimension tasks. When a "good" expert prior is incorporated, DynMeanBO consistently finds better solutions faster than the standard BO across different acquisition functions.

**"Good" expert prior.** Under the "good" expert prior, we compare $\pi$BO, ColaBO, and DynMeanBO on these two high-dimensional tasks. While $\pi$BO employs EI, ColaBO utilizes LogEI and MES, denoted as MCpi-LogEI and MCpi-MES in the figures. For a fair comparison, we evaluate DynMeanBO using the same acquisition functions—EI, LogEI, and MES. The results are shown in Figure 14. Across both tasks, all three prior-based methods (DynMeanBO, $\pi$BO, and ColaBO) consistently accelerate convergence under the "good" expert prior, and their performance is comparable.

We also compare the average per-iteration evaluation time of $\pi$BO, ColaBO, and DynMeanBO, as shown in Figure 15. As discussed in Section 5.2, DynMeanBO introduces negligible computational overhead compared to standard BO. The results further show that under the "good" expert prior, although all three methods achieve similarly strong optimization performance, DynMeanBO incurs noticeably lower computational cost compared to both $\pi$BO and ColaBO.

**"Bad" expert prior.** In practice, our expert prior is usually not perfectly accurate, but it rarely performs very poorly. In the few cases where the expert prior is indeed poor, we evaluate the robustness of DynMeanBO by comparing its performance with $\pi$BO and ColaBO under a "bad" expert prior. The results on the two high-dimensional tasks are shown in Figure 16, from which we can see that DynMeanBO exhibits strong robustness.

We also compare the average per-iteration evaluation time of $\pi$BO, ColaBO, and DynMeanBO under the "bad" expert prior, as shown in Figure 17.

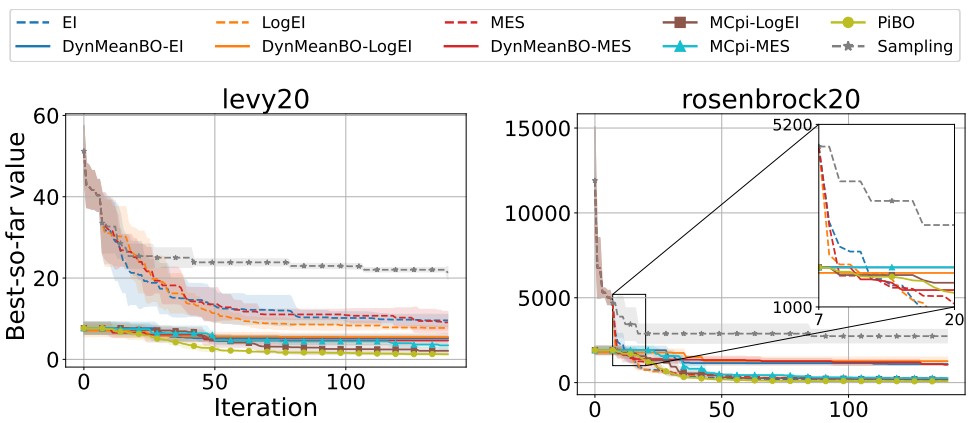

Figure 14: Performance on two high dimension tasks under a "good" expert prior. `DynMeanBO`, $\pi$`BO`, and `ColaBO` achieve comparable results.

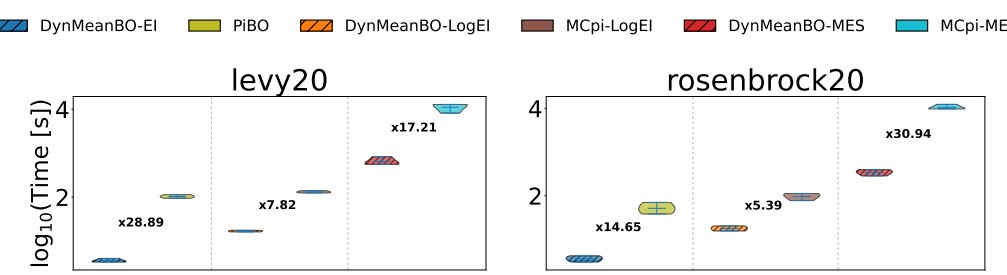

Figure 15: Per-iteration evaluation time ($\log_{10}$ scale) of `DynMeanBO`, $\pi$`BO`, and `ColaBO` on two high dimension tasks under the "good" expert prior setting.

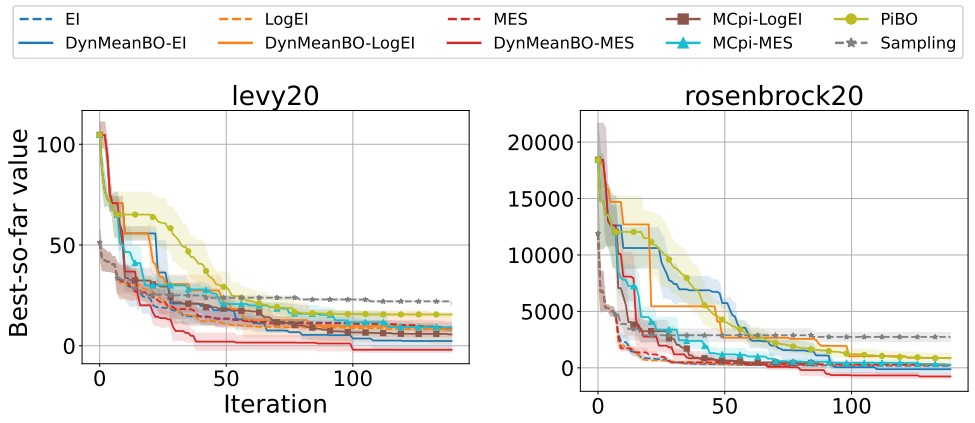

Figure 16: Performance on two high dimension tasks under a "bad" expert prior. `DynMeanBO` demonstrates strong robustness.

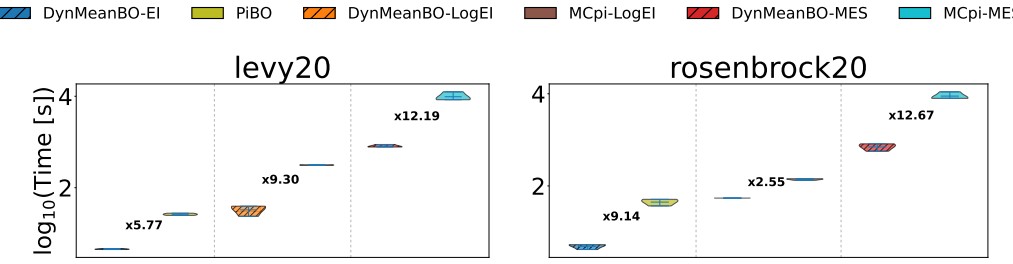

Figure 17: Per-iteration evaluation time ($\log_{10}$ scale) of `DynMeanBO`, $\pi$`BO`, and `ColaBO` on two high dimension tasks under the "bad" expert prior setting.

## J  SENSITIVITY ANALYSIS OF DECAY COEFFICIENT $\lambda$

When constructing the mean function, we define it as $m_n(\mathbf{x}) = \gamma_n m_{\text{prior}}(\mathbf{x}) + (1 - \gamma_n)\mu_0(\mathbf{x})$, $\gamma_n = \exp(-\lambda(n - N_0))$ ,where $m_{\text{prior}}(\mathbf{x}) = A \cdot \pi(\mathbf{x}) + B$. The hyperparameter $\lambda > 0$ controls the decay rate of the expert prior. In principle, a smaller $\lambda$ slows down the decay, allowing the expert prior to influence the optimization for a longer period and thereby reducing exploration in other regions. Conversely, a larger $\lambda$ accelerates the decay, diminishing the expert prior's influence earlier and encouraging broader exploration. To examine the effect of $\lambda$, we conducted an ablation study with $\lambda = 0.25, 0.5, 1.0$, and $2.0$ using EI as the acquisition function across eight benchmark tasks. In these experiments, we used the "good" expert prior. The results are shown in Figure 18.

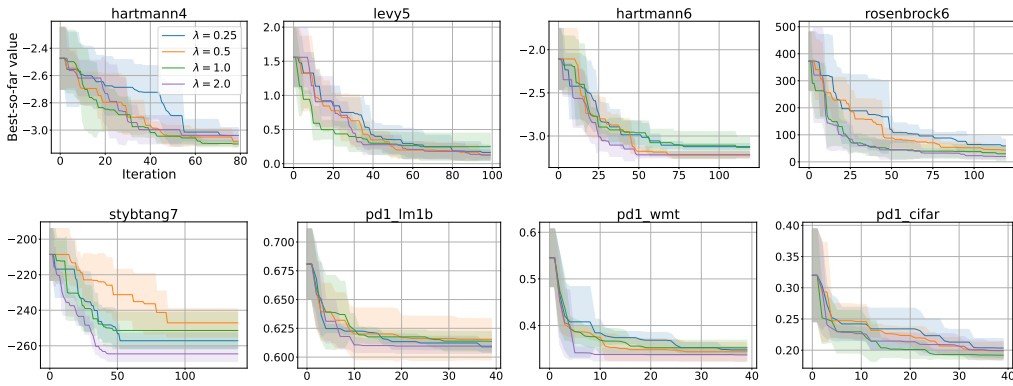

Figure 18: Ablation study of the parameter $\lambda$ under the "good" expert prior setting

From the experimental results, we observe a pattern consistent with the above analysis. A smaller $\lambda$ prolongs the influence of the expert prior, leading to less exploration in the early stages of optimization. If the expert prior is believed to be highly reliable, a smaller $\lambda$ is therefore preferable. In contrast, a larger $\lambda$ shortens the duration of the prior's influence and results in more exploration early on. Thus, if the expert prior is considered less reliable, choosing a larger $\lambda$ is more appropriate.

## K  SENSITIVITY ANALYSIS OF INITIALIZATION RATIO $\rho$

In our full algorithm (see Algorithm 1), we generate the initial points $N_0$ by combining samples from the expert prior distribution with Sobol sequences, which helps achieve a better trade-off between exploitation and exploration. The initialization ratio $\rho$ indicates the proportion of initial points drawn from the expert prior: a larger $\rho$ corresponds to more points sampled from the expert prior, while a smaller $\rho$ corresponds to fewer such points. To investigate the sensitivity to $\rho$, we conducted an ablation study with $\rho = 0, 0.4, 0.8$, and $1.0$ using EI as the acquisition function and the "good" expert prior across eight benchmark tasks. As a baseline, we used standard BO-EI. The experimental results are shown in Figure 19.

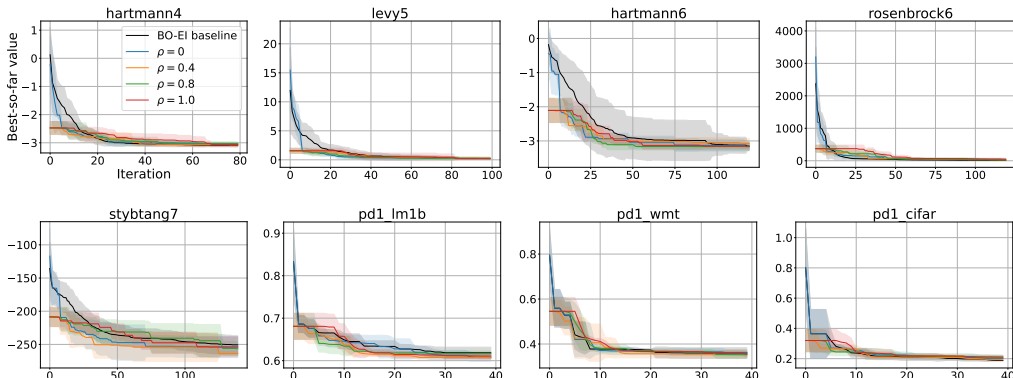

Figure 19: Ablation study of the parameter $\rho$ under the "good" expert prior setting

From the results of the ablation study, we observe that sampling entirely from the Sobol sequence ($\rho = 0$) performs worse than sampling some points from the expert prior ($\rho > 0$). This is because, in the initialization phase, if the expert prior is sufficiently reliable, points drawn from it tend to be closer to the optimum than those sampled from Sobol.

We also observe that sampling entirely from the expert prior ($\rho = 1$) performs worse than a combination of Sobol and expert prior sampling ($0 < \rho < 1$).

Finally, even when all initial points are sampled from Sobol ($\rho = 0$) (as can be seen in the Figure 19, where the blue and black lines start from the same position), `DynMeanBO-EI` still outperforms standard BO-EI, especially on the Hartmann4, Hartmann6, and Stybtang7 tasks. This is because, although both `DynMeanBO-EI` and BO-EI initialize entirely from Sobol when $\rho = 0$, `DynMeanBO` incorporates the expert prior into its mean function, which guides subsequent iterations toward better regions and enables faster convergence. These results demonstrate the effectiveness of integrating expert prior knowledge into the mean function.

Therefore, for the choice of the initialization ratio $\rho$, we recommend $0 < \rho < 1$. If the expert prior is sufficiently reliable, a larger $\rho$ is preferable, whereas if the expert prior is less reliable, a smaller $\rho$ is more appropriate.

## L    EXPERIMENTS UNDER DIFFERENT PRIOR STRENGTHS

In the main text, we compared a "good" expert prior (offset = 0.1 from the optimum, with a standard deviation equal to 0.2 times the length of the search-space interval) and a "bad" expert prior (offset = 0.7 from the optimum, with the same standard deviation). To more comprehensively characterize different prior conditions, we further examine strong expert priors, weak expert priors, and wrong expert priors, as well as different uncertainty levels with standard deviations set to $0.2, 0.4, 0.6,$ and $0.8$ times the search-space interval. The detailed configurations of the strong, weak, and wrong expert priors are provided in Table 2. Here, we still use the EI acquisition function; that is, this section presents an ablation study of different expert prior settings under `DynMeanBO-EI`. The experimental results are shown in Figure 20.

Table 2: Settings of strong, weak, and wrong expert priors.

| Expert Prior Type | Offset | Std (× search-space length) |
|---|---|---|
| Strong Expert Prior | 0.05 | 0.2 |
| Weak Expert Prior | 0.20 | 0.2 |
| Wrong Expert Prior | 0.90 | 0.2 |

From the experimental results, we observe that the variance of the expert prior has little effect on the optimization performance, whereas the quality of the expert prior has a significant impact. When a

strong expert prior is used, the `DynMeanBO` algorithm converges to the optimum almost immediately. With a weak expert prior, `DynMeanBO` performs slightly better than standard BO. In the case of a wrong expert prior, the initialization phase is initially misled, resulting in worse convergence compared to standard BO. However, due to the robustness of `DynMeanBO`, it eventually converges to a performance comparable to that of the original BO.

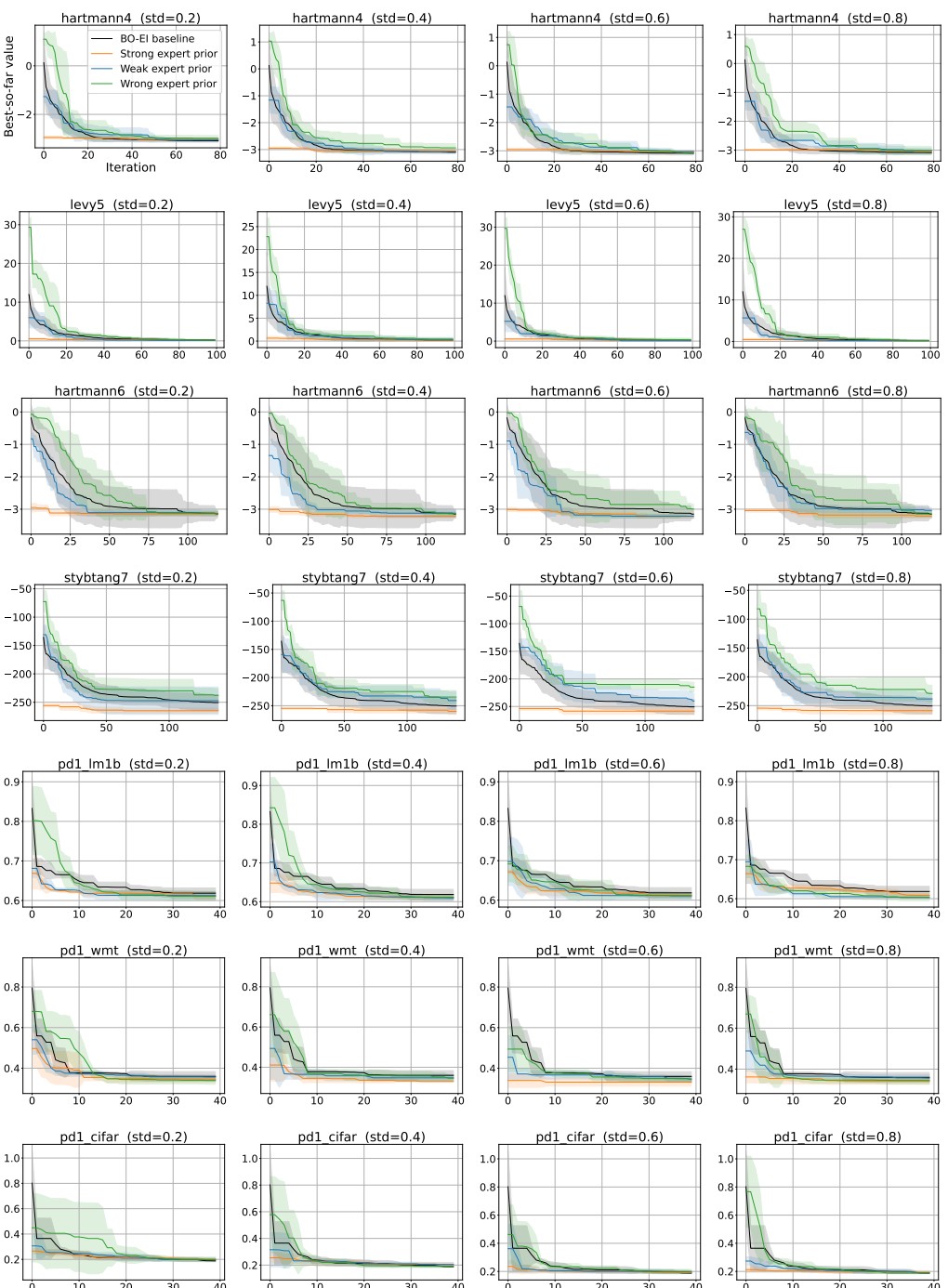

Figure 20: A comparative analysis of strong, weak, and wrong expert priors under varying prior variances.

## M    USE OF LARGE LANGUAGE MODELS

We employed the large language model GPT-4 to polish the manuscript. The process mainly involved: (1) checking for grammatical errors; (2) evaluating the appropriateness of voice usage; and (3) identifying and eliminating redundant expressions.

