# OpenReview forum: "Incorporating Expert Priors into Bayesian Optimization via Dynamic Mean Decay"
_ICLR.cc/2026/Conference — ICLR 2026 Poster_

### Official Review · Reviewer_xfoY · 2025-10-30

**Soundness:** 3
**Presentation:** 3
**Contribution:** 3
**Rating:** 6
**Confidence:** 4

**Summary:**

This paper introduces Dynamic Mean Bayesian Optimization (DynMeanBO) that incorporates expert priors into the Gaussian process mean function with a dynamic decay mechanism. Expert priors are modeled as probability distributions over the likely location of the global optimum. The dynamic decay mechanism is introduced to handle potential bias from bad priors. This approach is lightweight, compatible with arbitrary acquisition functions (AFs). Theoretical analysis provides convergence guarantees: DynMeanBO-EI and DynMeanBO-UCB. Experiments evaluate DynMeanBO on synthetic functions and hyperparameter optimization (HPO) tasks, using "good" and "bad" Gaussian priors. The proposed method accelerates convergence with informative priors, remains robust to misleading ones, and outperforms baselines like πBO and ColaBO.

**Strengths:**

1. The paper addresses a practical gap in prior-informed BO by integrating priors directly into the GP mean function in a principled way, rather than heuristically modifying AFs or kernels. The dynamic decay is a clever, simple mechanism that balances exploitation of priors early on with robustness later.
2. The paper provides solid convergence proof for EI and UCB, extending standard results to the dynamic mean setting.
3. The writing is clear. Figures are helpful, and appendices are referenced for details.

**Weaknesses:**

1. The proposed method relies exclusively on GPs, which may not extend easily to other surrogates like random forests or neural networks used in modern BO.
2. Parameters like the decay rate, scaling/shift, and initial points could be sensitive, but tuning guidance is minimal. In bad prior cases, if decay is too slow, bias might persist. Thus, empirical ablation would help.

**Questions:**

Please see the Weaknesses.

---

> ### Author Response · Authors · 2025-11-21
> **Response to Reviewer xfoY's Weaknesses and Questions**
>
> Dear reviewer xfoY,
>
> We sincerely thank you for taking the time to review our paper and for providing your valuable comments.
>
> **Weakness 1: The proposed method relies exclusively on GPs, which may not extend easily to other surrogates like random forests or neural networks used in modern BO.**
>
> We thank the reviewer for the insightful comments. Indeed, the current design of DynMeanBO is based on Gaussian Processes (GPs), as GPs provide a well-defined mean function, which allows us to incorporate expert priors in a theoretically grounded and analytically tractable manner.
> However, the core idea of DynMeanBO—dynamically adjusting the influence of expert priors during optimization—is conceptually general and not limited to GP models. In principle, this idea can be extended to other surrogate models, such as random forests or neural networks, by introducing analogous “prior bias” terms or adaptive regularization in their prediction components to achieve similar dynamic prior control. We consider this a promising direction for future research and have added a discussion of this extension in the Methods section (Section 4) of the revised manuscript.
>
> **Weakness 2: Parameters like the decay rate, scaling/shift, and initial points could be sensitive, but tuning guidance is minimal. In bad prior cases, if decay is too slow, bias might persist. Thus, empirical ablation would help.**
>
> This is an excellent suggestion, which will help us further improve our paper. In Appendix B, we provide a sensitivity analysis for parameters $A$ and $B$; in Appendix J, we analyze the sensitivity of the decay coefficient $\lambda$; and in Appendix K, we include an ablation study on the initialization rate $\rho$.
>
> ---
>
> Note: The additional experiments are still in progress. In the updated PDF, we have included the results that have been completed so far. Once all experiments are finished, we will update the PDF with the full results. Thank you for your patience and understanding.

---

> > ### Comment · Reviewer_xfoY · 2025-11-26
> >
> > I appreciate the authors’ revisions. I raise my confidence from 4 to 5.

---

> > > ### Author Response · Authors · 2025-11-27
> > > **Replying to Response to Reviewer xfoY's Comment**
> > >
> > > Dear reviewer xfoY,
> > >
> > > Thank you for the improved confidence score. We appreciate your assessment and will continue refining the details to further improve the paper.

---

### Official Review · Reviewer_udxX · 2025-10-30

**Soundness:** 4
**Presentation:** 4
**Contribution:** 2
**Rating:** 6
**Confidence:** 3

**Summary:**

The authors claim to contribute:
- A lightweight framework to incorporate expert priors into the GP mean function and decay it toward standard BO behavior, so that the resulting method can take advantage of a good prior but be robust to a bad one
- Convergence guarantees under EI and UCB acquisitions
- DynMeanBO performs comparably to existing methods that use a prior in good/bad prior case, but is much more computationally efficient because it only uses the prior in the GP mean function.

**Strengths:**

- Paper is well-organized and the main ideas and contributions concisely laid out. The background is thorough. The related works is thorough and motivates their method well. The method is simple and articulated clearly.
- DynMeanBO is substantially faster than MCpi (~10x?), and a little bit faster than PiBO. It would be good to know exactly how much faster it is than each method (perhaps write in the text or caption).
- DynMeanBO seems roughly comparable to PiBO and MCpi, though those methods seem to perform a little better in general, but is more computationally efficient (Fig. 4). It might be more informative to fix the total computation time to some budget, and then show regret for the more expensive methods, which may be able to take many fewer queries, with compute time as the x-axis. It seems generally better than PiBO and MCpi when a bad prior is given (Fig. 6). I’d also be curious to see what happens when a really bad prior is given (e.g., strong and in a place pretty far from the optimum) — does DynMeanBO still recover, or does it perform much worse than using regular BO?

**Weaknesses:**

- The prior is only incorporated into the GP mean function, though it seems desirable that it affect the covariance matrix too. It seems previous works have tried this and so far found it computationally expensive or unstable, though.
- The method seems relatively simple, which is not necessarily a bad thing, but I’m unsure if it constitutes a substantive contribution to the BO community.
- The regret results are not super impressive, though the method is much faster to run. It seems like this would be useful in certain BO cases, but it’s not totally clear to me how important this kind of work is, since BO typically deals with cases where querying the objective function is pretty expensive such that it’s often not a huge deal if it takes a while to choose the next point to evaluate. Some more discussion of the relative time for DynMeanBO vs. acquisition costs in real-world BO settings, compared to ColaBO/PiBO vs. acquisition costs would help elucidate the real-world relevance and applicability of this method.
- Fig.2: It’s pretty difficult to read the graphs because they’re so small. Maybe a log scale should be used for the y-axis. There primarily seems to be a performance improvement at early iterations, which makes sense because the prior just starts in the good region. However, on none of these problems is this gain held in later iterations.
- The method seems highly dependent on choosing a prior decay schedule, which should to some extent express how good the prior is expected to be anyway. It seems like it might be easier to just choose a weaker (more dispersed) prior instead. I can appreciate that they’ve chosen an exponential decay with only one hyperparam, $\lambda$, which seems to work reasonably well in the cases tested.
- I’m unsure how meaningful the theoretical results are in that they seem to rely on the fact that the mean will revert to having effectively no prior influence eventually, which seems like a trivial statement—if I do something for a finite number of iterations and then revert to a BO procedure that already has convergence guarantees, it’s unsurprising that this method also converges. That being said, I didn’t read the details super closely.

My recommendation would be weak accept--I think the paper and experiments are really nicely executed, but I'm unsure if the methodological contribution and results matter enough. I'm not super well-versed in the BO + priors subfield though.

*Some general feedback:*

Please make figure legends larger so they’re easier to read (e.g., Fig. 1). Also typo in Fig.1 — “Samples *form* prior” —> from. Some of the figures are generally too small / too many plots; please print out your paper and scale stuff so it’s readable in print.
Based on the algorithm description, I wouldn’t expect the runtime to be that much different from not using a prior, so I don’t think it’s that important to put this as Figure 3 — maybe a later figure, or supplemental? Figure 5 is more informative, maybe these two can be combined? It also might be helpful to make one big plot instead of many little plots, where the x-axis is the different problems, and each method is plotted as a line with error bars.

**Questions:**

I’d be curious to see what happens when a really bad prior is given (e.g., strong and in a place pretty far from the optimum) — does DynMeanBO still recover, or does it perform much worse than using regular BO?
What are the compute time factors for DynMeanBO vs. PiBO, ColaBO respectively? 10x faster? 2x faster? Specify for both good and bad prior experiments if they’re different. Also, some more discussion of the relative time for DynMeanBO vs. acquisition costs in real-world BO settings, compared to ColaBO/PiBO vs. acquisition costs would help elucidate the real-world relevance and applicability of this method.

---

> ### Author Response · Authors · 2025-11-21
> **Response to Reviewer udxX's Weaknesses 1-3**
>
> Dear reviewer udxX，
>
> We sincerely appreciate your thoughtful questions and detailed feedback, which we believe are extremely helpful for revising and improving our paper.
>
> **Weakness 1: The prior is only incorporated into the GP mean function, though it seems desirable that it affect the covariance matrix too. It seems previous works have tried this and so far found it computationally expensive or unstable, though.**
>
> Indeed, incorporating expert priors into the covariance  of a GP is appealing, as it allows the prior to influence the correlation structure between input points. However, as we noted in the Related Work section, the approach proposed in paper [1] has already explored a similar idea, and we also analyzed its limitations in our paper.
>
> Specifically, when the prior is injected into the kernel, each element of the kernel matrix K, i.e., k(x,x'), is affected by the prior. Since the kernel matrix underlies the entire BO process---including variance computation and covariance decomposition---any bias in the expert prior can propagate throughout K, distorting the variance estimation and potentially steering the optimization in the wrong direction. Moreover, complex kernel structures significantly increase computational cost (especially during matrix decomposition) and may lead to numerical instability, such as the kernel matrix becoming non-positive definite.
>
> In contrast, we found that injecting the expert prior solely into the mean function effectively achieves our goal: leveraging prior knowledge to guide exploration in the early stages---without introducing additional computational overhead or stability issues. Therefore, we opted not to incorporate the prior into the kernel.
>
> [1] Ramachandran A, Gupta S, Rana S, et al. Incorporating expert prior in Bayesian optimisation via space warping[J]. Knowledge-Based Systems, 2020, 195: 105663.
>
> **Weakness 2:  The method seems relatively simple, which is not necessarily a bad thing, but I’m unsure if it constitutes a substantive contribution to the BO community.**
>
> Indeed, our method is simple, which is by design. We avoided more complex approaches with limited benefit, aiming for an effective, robust, and broadly applicable solution that naturally incorporates expert priors into BO without added complexity.
>
> We believe this work provides a new perspective on effectively integrating expert priors into BO and can offer insights for related areas such as hyperparameter optimization (HPO). Therefore, we are confident that our study makes a meaningful contribution to the BO community and HPO, both in terms of practical utility and theoretical significance.
>
> **Weakness 3:  The regret results are not super impressive, though the method is much faster to run. It seems like this would be useful in certain BO cases, but it’s not totally clear to me how important this kind of work is, since BO typically deals with cases where querying the objective function is pretty expensive such that it’s often not a huge deal if it takes a while to choose the next point to evaluate. Some more discussion of the relative time for DynMeanBO vs. acquisition costs in real-world BO settings, compared to ColaBO/PiBO vs. acquisition costs would help elucidate the real-world relevance and applicability of this method.**
>
> We agree with the reviewer that when the number of evaluations is very large, the performance of our algorithm in terms of regret is not significantly better than that of standard BO. This is expected, as our theoretical analysis shows that the asymptotic convergence bounds of DynMeanBO are the same as those of standard BO, and thus their convergence rates are similar when the number of evaluations approaches infinity.
>
> However, BO is typically applied in scenarios where objective function evaluations are extremely expensive. In such cases, conducting a large number of evaluations is impractical; if evaluations were cheap, simple grid search or random search would suffice. Therefore, it is the convergence speed in the early stage under a limited evaluation budget that matters most. By incorporating expert priors, our algorithm can significantly accelerate convergence when the evaluation budget is limited.
> From our experimental results, we observe that when the expert prior is reasonably accurate, DynMeanBO achieves much faster performance improvement in the early evaluation stages compared to standard BO. This early-stage gain is precisely the phase that is most relevant in practical, high-cost optimization scenarios, and it is the main motivation for injecting expert priors into BO.
> Additionally, in the experimental section, we provide a more detailed comparison of the computational overhead of DynMeanBO, ColaBO, and PiBO, which further demonstrates the practical efficiency and applicability of our method.

---

> ### Author Response · Authors · 2025-11-21
> **Response to Reviewer udxX's Weaknesses 4-6**
>
> **Weakness 4: Fig.2: It’s pretty difficult to read the graphs because they’re so small. Maybe a log scale should be used for the y-axis. There primarily seems to be a performance improvement at early iterations, which makes sense because the prior just starts in the good region. However, on none of these problems is this gain held in later iterations.**
>
> We thank the reviewer for pointing out the readability issue in the figures and apologize for the inconvenience. To present the results more clearly, we have zoomed in on the important parts of the figures.
>
> As mentioned in our response to the previous comment, in practical BO scenarios, objective function evaluations are often extremely expensive and time-consuming, which limits the total number of evaluations. This makes the early stage of the optimization process the most critical. Theoretical analysis also indicates that when the number of evaluations is sufficiently large (i.e., in the later stage), DynMeanBO converges to performance comparable to standard BO. However, in real-world applications, evaluations often do not reach this late stage. Therefore, improvements in early-stage performance are most important, which is precisely where DynMeanBO demonstrates its main advantage.
>
> **Weakness 5: The method seems highly dependent on choosing a prior decay schedule, which should to some extent express how good the prior is expected to be anyway. It seems like it might be easier to just choose a weaker (more dispersed) prior instead. I can appreciate that they’ve chosen an exponential decay with only one hyperparam, $\lambda$, which seems to work reasonably well in the cases tested.**
>
> We fully agree with the reviewer that the decay coefficient $\lambda$ affects the influence of the expert prior in BO. A larger $\lambda$ means that the prior primarily affects the early evaluations, while a smaller $\lambda$ extends the guidance of the prior over more iterations. This parameter is flexible and can be adjusted according to the trustworthiness of the expert prior: if the prior is considered highly reliable for a given task, we can set $\lambda$ smaller; if the prior is less reliable, a larger $\lambda$ is preferable.
>
> To address the reviewer’s concern about parameter sensitivity, we provide ablation studies and a sensitivity analysis for $\lambda$ in Appendix J.
>
> **Weakness 6:  I’m unsure how meaningful the theoretical results are in that they seem to rely on the fact that the mean will revert to having effectively no prior influence eventually, which seems like a trivial statement—if I do something for a finite number of iterations and then revert to a BO procedure that already has convergence guarantees, it’s unsurprising that this method also converges. That being said, I didn’t read the details super closely.**
>
> We thank the reviewer for their attention to the theoretical analysis in our paper. The convergence guarantees of standard BO are derived under the assumption of a zero mean function. Since DynMeanBO incorporates the expert prior into the mean function, the mean function becomes non-zero, which might raise concerns about whether convergence is affected. Our theoretical analysis ensures that DynMeanBO does not diverge and that its asymptotic convergence is consistent with standard BO. This is an important property, as it guarantees that introducing expert priors does not compromise the theoretical soundness of the algorithm.
>
> **Some general feedback: Please make figure legends larger so they’re easier to read (e.g., Fig. 1). Also typo in Fig.1 — “Samples form prior” —> from. Some of the figures are generally too small / too many plots; please print out your paper and scale stuff so it’s readable in print. Based on the algorithm description, I wouldn’t expect the runtime to be that much different from not using a prior, so I don’t think it’s that important to put this as Figure 3 — maybe a later figure, or supplemental? Figure 5 is more informative, maybe these two can be combined? It also might be helpful to make one big plot instead of many little plots, where the x-axis is the different problems, and each method is plotted as a line with error bars.**
>
> We sincerely thank the reviewer for their careful observations. We have corrected the typographical error in Figure 1 and enlarged the legend and relevant fonts to improve readability. Following your valuable suggestion, we have also reorganized the layout of the figures to make the overall presentation clearer and easier to understand.
> As you pointed out, Figure 3 is not strictly necessary in the main text. Since injecting the expert prior into the mean function adds negligible computational overhead according to the algorithm design, we have moved this figure to the Appendix G.

---

> ### Author Response · Authors · 2025-11-21
> **Response to Reviewer udxX's Questions**
>
> **Questions:  I’d be curious to see what happens when a really bad prior is given (e.g., strong and in a place pretty far from the optimum) — does DynMeanBO still recover, or does it perform much worse than using regular BO? What are the compute time factors for DynMeanBO vs. PiBO, ColaBO respectively? 10x faster? 2x faster? Specify for both good and bad prior experiments if they’re different. Also, some more discussion of the relative time for DynMeanBO vs. acquisition costs in real-world BO settings, compared to ColaBO/PiBO vs. acquisition costs would help elucidate the real-world relevance and applicability of this method.**
>
> In Appendix L, we studied three substantially different expert priors: strong expert prior, weak expert prior, and wrong expert prior. The wrong expert prior can be considered a highly misleading prior, as its optimum is located far from the true optimum. Our experimental results show that DynMeanBO exhibits strong robustness: even when faced with such an extremely poor expert prior, it is able to rapidly converge to performance comparable to standard BO.
> In our experiments, the computational time figures highlight how many times faster DynMeanBO is compared to PiBO and ColaBO and we provide corresponding discussion in the text.
>
> ---
>
> Note: The additional experiments are still in progress. In the updated PDF, we have included the results that have been completed so far. Once all experiments are finished, we will update the PDF with the full results. Thank you for your patience and understanding.

---

> ### Author Response · Authors · 2025-11-27
> **Response to Reviewer udxX**
>
> Dear reviewer udxX，
>
> Thank you very much for raising many valuable questions and pointing out errors that we had previously overlooked. Your suggestions and comments have greatly helped us improve and refine this paper. The latest version of the PDF has been uploaded, and we would be very glad to discuss further if you have time.

---

### Official Review · Reviewer_ZGXS · 2025-11-01

**Soundness:** 2
**Presentation:** 3
**Contribution:** 3
**Rating:** 4
**Confidence:** 3

**Summary:**

The paper introduces DynMeanBO, a Bayesian optimization framework that incorporates expert prior knowledge into a GP mean function and gradually reduces that influence over time using an exponential decay factor. This dynamic mean decay allows the optimizer to exploit useful expert beliefs early on but prevents long-term bias from incorrect priors, making the method computationally light, acquisition-function agnostic, and preserving standard convergence guarantees. Empirical results across synthetic functions and hyperparameter tuning tasks show that DynMeanBO matches or outperforms prior-informed BO methods while being more robust and nearly as fast as standard BO.

**Strengths:**

1.	Integrating prior distributions into the GP mean with a decaying coefficient is conceptually straightforward and compatible with arbitrary acquisition functions. It avoids the complexity of prior-specific acquisition designs.
2.	By decaying the influence of the expert mean, the method can recover from misleading prior beliefs. Theoretical analysis establishes that DynMeanBO does not sacrifice long-term performance despite using potentially biased priors. The authors prove that under standard assumptions (e.g. the objective lies in the RKHS of the GP kernel), DynMeanBO with EI or UCB achieves the same asymptotic convergence rate and regret bounds as standard BO.
3.	The approach is modular and compatible with a broad set of acquisition functions, which avoids the complexity of prior-specific acquisition designs.

**Weaknesses:**

1.	Most experiments involve low  to medium dimensional tasks and a specific set of deep-learning hyperparameter benchmarks; it remains unclear how DynMeanBO performs in very high-dimensional or multi-fidelity scenarios, or on domains where priors are scarce.
2.	The paper did not provide detailed analysis on how sensitive the method is to mismatched prior modes or variances, which would help practitioners understand risk when experts provide only coarse or biased priors.
3.	Although mixtures of Gaussians are considered, the framework implicitly assumes that expert priors can be parameterized easily and effectively, which may not hold in many real-world domains.
4.	DynMeanBO introduces new tuning parameters (for the prior mean and its decay) that are set heuristically and not thoroughly examined. The GP mean uses a scaled version of the expert’s prior distribution $m_{\text{prior}}(x) = A,\pi(x) + B$, but the paper fixes $A=1$ and $B=0$ “for simplicity” without discussing how these might affect results. Likewise, the decay schedule $\gamma_n=\exp(-\lambda (n-N_0))$ uses a fixed rate $\lambda=1$ in all experiments, and 40% of initial samples are drawn from the prior (a fixed ratio). There is no ablation study or guidance on choosing these hyperparameters. This raises questions: e.g. if $\lambda$ is too small (slow decay), a bad prior might mislead longer, whereas if too large, a good prior’s benefit might vanish prematurely. The robustness and performance of DynMeanBO could be sensitive to these settings, but the paper provides only a single fixed configuration. Without exploring different decay rates or prior weightings, it’s unclear how one would tune DynMeanBO in practice when the reliability of the expert prior is unknown.

**Questions:**

1. How sensitive is DynMeanBO’s performance to the choice of decay rate $\lambda$ and the number of initial prior-guided points $N_0$? The experiments fixed $\lambda=1$ and used a 40% prior-based initialization without ablation. Would a slower or faster decay significantly change the outcomes?
2. Beyond the specific “bad prior” tested (a single offset Gaussian), are there other failure modes to be aware of? For example, what if the expert prior is not just slightly off in location but completely adversarial (e.g. concentrated in a region of very low function value)? In such extreme cases, does the hybrid initialization (mixing Sobol points) and decay suffice to guarantee recovery, or could there be situations where DynMeanBO still performs worse than standard BO for a while? A related point is that how does observation noise affect the trust in the prior? If the objective evaluations are noisy, one might need more iterations to “override” a misleading prior.

---

> ### Author Response · Authors · 2025-11-21
> **Response to Reviewer ZGXS's Weaknesses**
>
> Dear reviewer ZGXS，
>
> We sincerely thank you for taking the time to review our paper and for providing constructive and insightful comments, which have greatly helped us improve the clarity and quality of our work.
>
> **Weakness1：Most experiments involve low to medium dimensional tasks and a specific set of deep-learning hyperparameter benchmarks; it remains unclear how DynMeanBO performs in very high-dimensional or multi-fidelity scenarios, or on domains where priors are scarce.**
>
> Indeed, most prior works in this field conduct experiments on low- to medium-dimensional tasks, as these are the most common in practice. We fully agree that it is important to evaluate DynMeanBO on higher-dimensional tasks as well. Accordingly, we have added experiments on two 20-dimensional tasks—Levy (20D), and Rosenbrock (20D)—with results presented in Appendix I.
>
> Since DynMeanBO is based on the BO framework, it can naturally be applied to multi-fidelity tasks within the BO framework; however, due to space and time constraints, we have not included such experiments in this version. Regarding scenarios where expert priors are scarce, the standard BO method can be directly applied. Nevertheless, in many real-world tasks, expert priors are often available (as discussed in Section 2.4), which motivates our work of integrating expert knowledge into BO.
>
> **Weakness 2： The paper did not provide detailed analysis on how sensitive the method is to mismatched prior modes or variances, which would help practitioners understand risk when experts provide only coarse or biased priors.**
>
> In fact, our paper already demonstrates that DynMeanBO is quite robust when the expert prior is “bad.”  To more clearly illustrate this phenomenon, Appendix L presents a comparative analysis of DynMeanBO performance under different prior variance settings, when the expert prior is strong, weak, or wrong. The results demonstrate the specific impact of different combinations of prior strength and variance on the optimization process.
>
> **Weakness 3：Although mixtures of Gaussians are considered, the framework implicitly assumes that expert priors can be parameterized easily and effectively, which may not hold in many real-world domains.**
>
> In fact, expert priors can be easily expressed as probability distributions, since we only need an approximate shape that reflects the prior knowledge. In this paper, we illustrate expert priors using two representative examples: a single Gaussian distribution and a Gaussian mixture distribution, but in principle, any probability distribution can be used. For instance, one can construct an arbitrary function h(x), where the location of the optimum of h(x) represents our prior belief about the task’s optimum, and then define the prior as π(x) = h(x) / ∫ h(x) dx. Further details on expert priors are provided in Section 2.4  and Appendix A of the paper.
>
> **Weakness 4: DynMeanBO introduces new tuning parameters (for the prior mean and its decay) that are set heuristically and not thoroughly examined. The GP mean uses a scaled version of the expert’s prior distribution $m_{\text{prior}} = A\pi(x) + B$, but the paper fixes $A=1$ and $B=0$ “for simplicity” without discussing how these might affect results. Likewise, the decay schedule $\gamma_n=\exp(-\lambda(n-N_0))$ uses a fixed rate $\lambda=1$  in all experiments, and 40% of initial samples are drawn from the prior (a fixed ratio). There is no ablation study or guidance on choosing these hyperparameters. This raises questions: e.g. if  $\lambda$  is too small (slow decay), a bad prior might mislead longer, whereas if too large, a good prior’s benefit might vanish prematurely. The robustness and performance of DynMeanBO could be sensitive to these settings, but the paper provides only a single fixed configuration. Without exploring different decay rates or prior weightings, it’s unclear how one would tune DynMeanBO in practice when the reliability of the expert prior is unknown.**
>
> We thank the reviewer for raising this important point, which greatly helps to strengthen and improve our work. Since we had not previously conducted detailed ablation experiments on these hyperparameters, this may have caused some confusion for the readers. Following your suggestion, we have now performed a sensitivity analysis for all these hyperparameters.
>
> The effects of parameters $A$ and $B$ are explained in Section 4.1 and Appendix B, with detailed ablation experiments provided in Appendix B. The decay coefficient $\lambda$ controls the influence of the expert prior: as you noted, a smaller $\lambda$ allows the prior's effect to persist longer, whereas a larger $\lambda$ encourages more exploration during optimization. We provide a sensitivity analysis of $\lambda$ in Appendix J. Additionally, we have conducted a sensitivity analysis for the initial sampling ratio $\rho$, which can be found in Appendix K.

---

> ### Author Response · Authors · 2025-11-21
> **Response to Reviewer ZGXS's Questions**
>
> **Question1: How sensitive is DynMeanBO’s performance to the choice of decay rate  $\lambda$ and the number of initial prior-guided points  $N_0$? The experiments fixed $\lambda=1$  and used a 40% prior-based initialization without ablation. Would a slower or faster decay significantly change the outcomes?**
>
> Based on your suggestion, we conducted an ablation study on the decay rate $\lambda$ in Appendix J and on the initialization ratio $\rho$ in Appendix K. Through our analysis and empirical evaluation, we observe that when the expert prior is considered reliable, using a smaller decay rate $\lambda$ and a larger initialization ratio $\rho$ allows more effective utilization of expert knowledge. Conversely, when the expert prior is uncertain or potentially misleading, a larger decay rate $\lambda$ and a smaller initialization ratio $\rho$ help mitigate the negative impact of an incorrect prior. Please refer to Appendices J and K for more details.
>
> **Question2:  Beyond the specific “bad prior” tested (a single offset Gaussian), are there other failure modes to be aware of? For example, what if the expert prior is not just slightly off in location but completely adversarial (e.g. concentrated in a region of very low function value)? In such extreme cases, does the hybrid initialization (mixing Sobol points) and decay suffice to guarantee recovery, or could there be situations where DynMeanBO still performs worse than standard BO for a while? A related point is that how does observation noise affect the trust in the prior? If the objective evaluations are noisy, one might need more iterations to “override” a misleading prior.**
>
> To better investigate this issue, we conducted an additional study in Appendix L, where we evaluated DynMeanBO under three substantially different expert prior settings: Strong Expert Prior, Weak Expert Prior, and Wrong Expert Prior. In practical applications, severely misleading expert priors usually manifest as systematic bias in the location or scale of the prior, rather than being uniformly adversarial across the entire search space. Therefore, instead of constructing a “fully adversarial” prior, we focus on a representative and practically meaningful failure mode—the Wrong Expert Prior, whose probability mass is concentrated far away from the true optimum. This setting reflects realistic scenarios where expert intuition is severely biased or outdated.
>
> As reported in Appendix L, even under this highly misleading prior, DynMeanBO is able to rapidly correct the initial bias and converge to performance comparable to standard BO, demonstrating strong robustness. While more pathological adversarial priors are theoretically possible, they are rarely encountered in practice. We consider investigation of such extreme adversarial cases to be a valuable direction for future work.
>
> In fact, all seven synthetic benchmark functions we evaluated (Hartmann4, Levy5, Hartmann6, Rosenbrock6, Stybtang7, Levy20, and Rosenbrock20) contain observation noise. In principle, when the objective function is noisy, DynMeanBO may require more evaluation rounds than in the noiseless case to overcome an initially misleading prior and converge to near-optimal performance. This is consistent with the results observed in Appendix L: even in the presence of noise, DynMeanBO can quickly correct the initial bias and achieve final performance comparable to standard BO.
>
> ---
>
> Note: The additional experiments are still in progress. In the updated PDF, we have included the results that have been completed so far. Once all experiments are finished, we will update the PDF with the full results. Thank you for your patience and understanding.

---

> ### Author Response · Authors · 2025-11-27
> **Response to Reviewer ZGXS**
>
> Dear reviewer ZGXS，
>
> Thank you for your valuable comments. We have made corresponding revisions based on your suggestions, and we hope the work is now more sound. The latest version of the manuscript (PDF) is now available. We welcome any further discussion or comments, and would be glad to engage in further communication if needed.

---

### Author Response · Authors · 2025-11-30
**Rebuttal Summary**

Dear Area Chair,

We sincerely appreciate the thoughtful and constructive feedback from all three reviewers. Their suggestions have greatly helped us refine and strengthen the paper. In response, we conducted additional experiments, improved the clarity and completeness of the manuscript, and have provided an updated version.

For your convenience, we provide a concise summary of the modifications and newly added materials below:

1. We enlarged the font size in Figure 1 and corrected the typo `form` to `from`. For the experimental figures (Figures 2, 3, 5, 13, and 14), we added zoomed-in views to highlight the key regions. In the runtime comparison figures (Figures 4, 12, 15, and 17), we further annotated the speed-up ratio to clearly show how many times faster each method is. These adjustments make the visual results more explicit and easier to interpret.

2. For the reviewers' concerns regarding the expert prior distribution, we have provided further clarification and explanation in the corresponding sections of the revised manuscript. Meanwhile, we reorganized relevant content for better logical flow. These revisions address the reviewers’ questions and significantly improve readability.

3. In Appendix B, we added sensitivity analysis experiments for parameters $A$ and $B$.

4. In Appendix I, we included additional high-dimensional experiments on two 20-dimensional tasks (Levy20 and Rosenbrock20).

5. In Appendix J, we added sensitivity analysis experiments for the decay coefficient $\lambda$.

6. In Appendix K, we added sensitivity analysis experiments for the initialization ratio $\rho$.

7. In Appendix L, we further included experiments under strong, weak, and wrong expert priors with different prior variances.

---

### Meta-Review · Area_Chair_XNTC · 2026-01-05

**Summary:**

The reviewers raised concerns on (1) possibilities of extension to more challenging BO settings, (2) missing analysis on sensitivity of hyperparameters, (3) effect of bad priors, (4) thin theoretical results, and (5) presentation of figures and some other minor ones. As a paper incorporating the expert priors into the BO framework, I don't think it's very important for this work to cover high-d, multi-fidelity, and non-GP settings. For other concerns, they are found to have been addressed by either the author rebuttal or the paper revision, so there should be a unanimous decision towards accept if all reviewers could engage with the discussion, therefore, I recommend accept.

**Reviewer Concerns:**

The reviewers raised concerns on (1) possibilities of extension to more challenging BO settings, (2) missing analysis on sensitivity of hyperparameters, (3) effect of bad priors, (4) thin theoretical results, (5) presentation of figures, and some other minor ones. All concerns are found to have been addressed.

**Reviewer Scores:**

Reviewer xfoY has raised their confidence score from 4 to 5 after reading the author rebuttal, showing continuous strong support for weak accept. Unfortunately, Reviewers ZGXS and udxX didn't reply to the author rebuttal, but given the detailed author rebuttal and the revision of the paper, all concerns are found to be have been addressed so I think they might raise their scores, leaning towards weak accept / accept.

---

### Decision · Program_Chairs · 2026-01-26

Accept (Poster)